# DeRaDiff: Denoising Time Realignment of Diffusion Models

**Ratnavibusena Don Shahain Manujith**[*], **Teoh Tze Tzun**[*], **Kenji Kawaguchi**, **Yang Zhang**
National University of Singapore
`{shahain,teoh.tze.tzun,yangzhang}@u.nus.edu`
`kenji@nus.edu.sg`

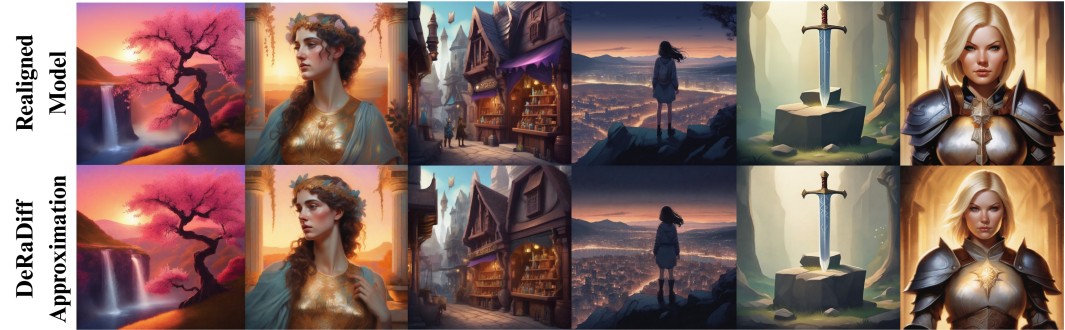

Figure 1: **DeRaDiff re-approximates a model aligned from scratch.** Top row consists of images generated by an SDXL model aligned from scratch at $\beta = 5000$ KL regularization strength. Bottom row consists of images obtained via DeRaDiff sampling via an anchoring SDXL model aligned at a KL regularization strength of $\beta = 2000$ with no further retraining.

## ABSTRACT

Recent advances align diffusion models with human preferences to increase aesthetic appeal and mitigate artifacts and biases. Such methods aim to maximize a conditional output distribution aligned with higher rewards whilst not drifting far from a pretrained prior. This is commonly enforced by KL (Kullback–Leibler) regularization. As such, a central issue still remains: how does one choose the right regularization strength? Too high of a strength leads to limited alignment and too low of a strength leads to "reward hacking". This renders the task of choosing the correct regularization strength highly non-trivial. Existing approaches sweep over this hyperparameter by aligning a pretrained model at multiple regularization strengths and then choose the best strength. Unfortunately, this is prohibitively expensive. We introduce *DeRaDiff*, a *denoising-time realignment procedure* that, after aligning a pretrained model once, modulates the regularization strength *during sampling* to emulate models trained at other regularization strengths—*without any additional training or fine-tuning*. Extending decoding-time realignment from language to diffusion models, DeRaDiff operates over iterative predictions of continuous latents by replacing the reverse-step reference distribution by a geometric mixture of an aligned and reference posterior, thus giving rise to a closed-form update under common schedulers and a single tunable parameter, $\lambda$, for on-the-fly control. Our experiments show that across multiple text–image alignment and image-quality metrics, our method consistently provides a strong approximation for models aligned entirely from scratch at different regularization strengths. Thus, our method yields an efficient way to search for the optimal strength, eliminating the need for expensive alignment sweeps and thereby substantially reducing computational costs. The official implementation is available at `github.com/itsShahain/DeRaDiff`.

---

[*]Indicates equal contribution.
Subject matter correspondence to: Shahain Manujith <shahain@u.nus.edu>

## 1 INTRODUCTION

Text-to-image (T2I) diffusion models (Ho et al. (2020); Rombach et al. (2022)) now underpin state-of-the-art image generation. Sampling has been made efficient by techniques such as classifier-free guidance and latent diffusion, unlocking applications like style transfer, image-to-image translation, and inpainting (Dhariwal & Nichol (2021); Saharia et al. (2022)). Most modern systems are trained in two stages: (i) pretraining, which optimizes the diffusion objective on large-scale data; and (ii) alignment, which adapts behavior to tasks or human preferences via supervised fine-tuning (SFT) (Lee et al. (2023)) or reinforcement learning (Black et al. (2023), Clark et al. (2023) ).

A persistent challenge in alignment is balancing adaptation with fidelity to the pretrained prior. This trade-off is typically controlled by a proximity penalty—most commonly a Kullback–Leibler (KL) divergence—between the aligned and reference distributions. The associated regularization strength is pivotal: if too strong, the model under-adapts; if too weak, it drifts and risks reward hacking (Amodei et al. (2016); Stiennon et al. (2020); Bai et al. (2022); Lewis et al. (2020) ). Unfortunately, identifying the right hyperparameter generally requires expensive sweeps that are prohibitive for large diffusion models (Ho et al. (2020); Rombach et al. (2022)).

To this end, we propose **DeRaDiff**, a *denoising-time realignment procedure*. In the context of language modeling, *realignment* is defined as the post-hoc adjustment of the regularization strength $\beta$—effectively modulating the proximity to the reference model—by geometrically mixing the reference and aligned distributions at inference time (Liu et al. (2024)). While this enables LLMs to vary alignment intensity via discrete logit manipulation, applying this principle to generative art presents a distinct challenge: diffusion models do not output single-step probabilities over a finite vocabulary, but rather operate via the iterative denoising of *continuous* latents. Our key insight is a derivation of a tractable, closed form formula for the geometric mixture of a reference and aligned diffusion models' distribution that is parameterized by $\lambda$ that adjusts the effective regularization strength relative to the aligned model's regularization strength, $\beta$. Crucially, $\lambda$ is tunable *on-the-fly* during inference.

As such, the closed-form update is presented in theorem 1 for the realigned reverse process, providing both a theoretical basis and an efficient implementation (see Algorithm 1). Quantitative (Section 5) and qualitative results (Figure 8, Figure 3) show that DeRaDiff preserves downstream performance while obviating retraining. Moreover, we achieve substantial compute savings, as described in section 6. Our contributions can be summarized as three-fold:

- A theoretical extension of decoding-time realignment to diffusion processes, yielding a closed-form stepwise realignment posterior integrated into the reverse diffusion process.
- **DeRaDiff**, a denoising-time realignment method that approximates models aligned at different regularization strength *without additional training* by modulating alignment during sampling.
- Experimental evidence that DeRaDiff enables precise control of alignment strength and accelerates RLHF-style hyperparameter exploration, substantially reducing compute while preserving downstream performance.

## 2 RELATED WORK

**Alignment of diffusion models.** A growing body of work aligns diffusion models using preference signals or task rewards, including DDPO (Black et al., 2023) , DRaFT (Clark et al., 2023) , DPOK (Fan et al., 2023), AlignProp (Prabhudesai et al., 2023) , and Diffusion DPO (Wallace et al., 2023) . These methods chiefly study the effectiveness and training efficiency of alignment procedures. Central to their stability is the choice of regularization strength toward a reference model: insufficient regularization permits distributional drift and reward hacking, whereby models score high rewards but fail on the intended task. (Amodei et al. (2016); Stiennon et al. (2020); Bai et al. (2022); Lewis et al. (2020) ).

**Decoding-time alignment of sampling distributions.** To avoid retraining for each task or preference setting, recent work has considered decoding-time control of the sampling distribution. One line of work leverages unconditionally pretrained diffusion models together with pretrained neural

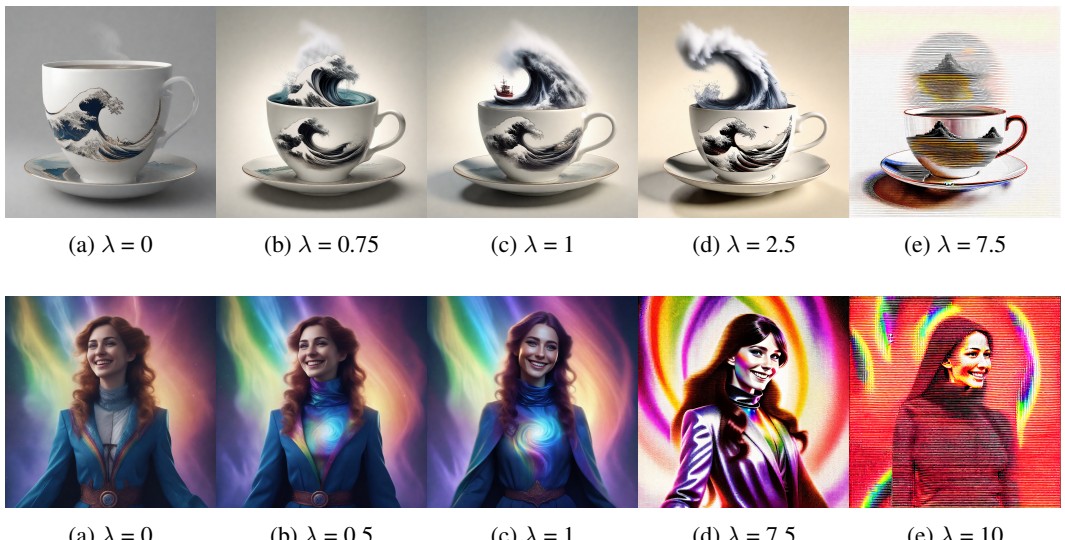

| (a) $\lambda = 0$ | (b) $\lambda = 0.75$ | (c) $\lambda = 1$ | (d) $\lambda = 2.5$ | (e) $\lambda = 7.5$ |

| (a) $\lambda = 0$ | (b) $\lambda = 0.5$ | (c) $\lambda = 1$ | (d) $\lambda = 7.5$ | (e) $\lambda = 10$ |

Figure 3: **On-the-fly modulation with DeRaDiff.** Applied to SDXL (Podell et al., 2023), DeRaDiff adjusts alignment at inference via a scalar $\lambda$. Increasing $\lambda$ decreases the effective regularization (more alignment to human preferences) and increases the aesthetic quality and increases prompt adherence as expected, while maintaining $\lambda \in [0, 1]$. (Top: *"Typhoon in a teacup..."*, bottom: *"A smiling beautiful sorceress..."*). However, increasing $\lambda$ beyond 1 pushes the model beyond the aligned regime, resulting in degradation in aesthetics and inducing reward-hacking-like artifacts as is expected for models trained on too low a regularization strength.

networks to enable diverse conditional generation tasks (He et al. (2024)). Another line of work employs Sequential Monte Carlo to sample from reward-aligned target distributions at inference-time (Kim et al. (2025); Wu et al. (2023a)). While effective, these approaches generally do not exploit the presence of a *conditional* model that has undergone alignment.

**Decoding-time realignment in language models.** In language modeling, Liu et al. (2024) introduced decoding-time realignment, offering a theoretical framework and empirical validation for decoding-time realignment of *discrete* next-token distributions. Our work differs by developing an analogous realignment mechanism for *continuous* diffusion trajectories, adapting realignment to the iterative denoising process and thereby enabling inference-time control of regularization strength without additional alignment for diffusion models.

**Decoding-time realignment in diffusion models.** Diffusion Blend (Cheng et al., 2025) introduced a decoding-time realignment for diffusion models under the score-based SDE (Song et al., 2020) paradigm. Our work differs by providing a decoding-time realignment approach under the DDPM paradigm (Sohl-Dickstein et al., 2015). Although (Karras et al., 2022) established a theoretical equivalence between DDPM and SDE paradigms, our approach establishes the theoretical foundation for an *exact* closed-form Gaussian update on the stepwise realigned posterior under mild assumptions. To the best of our knowledge, our work is the first to introduce realignment under the DDPM paradigm and provide a theoretical foundation for the stepwise posterior.

## 3 BACKGROUND

**Diffusion models.** We follow the common latent diffusion formulation (Rombach et al., 2022): given noise schedule parameters $\{\alpha_t, \sigma_t\}_{t=0}^T$, a denoising diffusion model (Ho et al. (2020), Sohl-Dickstein et al. (2015)) defines a Markovian reverse process. Here,

$$p_\theta(x_{0:T}) = \prod_{t=1}^T p_\theta(x_{t-1}|x_t), \qquad p_\theta(x_{t-1}|x_t) = \mathcal{N}\left(x_{t-1}; \mu_\theta(x_t, t, c),\ \sigma_{t|t-1}^2 \frac{\sigma_{t-1}^2}{\sigma_t^2} I\right), \qquad (1)$$

**KL-regularized RL fine-tuning.** Following Jaques et al. (2017) and Jaques et al. (2020), alignment is commonly cast as reward maximization with a KL penalty to keep the fine-tuned model near a pretrained reference:

$$\max_{p_\theta} \mathbb{E}_{c\sim\mathcal{D}_c, x_0\sim p_\theta(x_0|c)}[r(c, x_0)] - \beta\mathbb{D}_{\text{KL}}[p_\theta(x_0|c)\|p_{\text{ref}}(x_0|c)] \tag{2}$$

Here, $\mathcal{D}_c$ is a distribution of prompts and $\beta > 0$ controls the trade-off between reward and proximity to the reference model. Sweeping $\beta$ is the standard way to find the desired alignment strength but is computationally expensive; our method provides an inference-time tool to cheaply explore this space for Diffusion Models. The unique global optimum of Equation 2 for the discrete case was explored by Ziegler et al. (2019), Korbak et al. (2022), Rafailov et al. (2023). We provide a natural extension of the unique global optimum for the continuous case (see appendix A.2 for details):

$$p_\theta^*[\beta](x_0|c) = \frac{p_{\text{ref}}(x_0|c)e^{\frac{1}{\beta}r(c,x_0)}}{\int p_{\text{ref}}(x_0'|c)e^{\frac{1}{\beta}r(c,x_0')}\,dx_0'} \tag{3}$$

**Realignment at decoding time** Decoding-time realignment (Liu et al. (2024)) blends a reference and an aligned model at sampling time. Extending this idea to diffusion models requires handling continuous per-step posteriors rather than discrete next-token distributions. In the next section, we derive a closed-form per-step Gaussian interpolation and give a complete sampling procedure (algorithm 1). Full technical derivations are provided in appendix A.

## 4 METHOD

### 4.1 REALIGNMENT & STEPWISE APPROXIMATION

We follow the formulation of decoding-time realignment (Liu et al. (2024)) which expresses the *realigned* model as a geometric mixture of the reference and aligned densities. Concretely, the full-sample posterior is given by (see appendix A.1 for details):

$$p_\theta^*[\beta/\lambda](x_0\,|\,c) = \frac{p_{\text{ref}}(x_0\,|\,c)^{1-\lambda}\ p_\theta^*[\beta](x_0\,|\,c)^\lambda}{\int p_{\text{ref}}(x_0'\,|\,c)^{1-\lambda}\ p_\theta^*[\beta](x_0'\,|\,c)^\lambda dx_0'}, \tag{4}$$

which is the normalized version of $p_{\text{ref}}^{1-\lambda}\,p_\theta^*[\beta]^\lambda$. Direct evaluation of Equation 4 is intractable for diffusion models as it requires marginalizing all intermediate latents. We therefore apply a *stepwise denoising approximation* and apply the same geometric mixture of the densities at each step:

$$\hat{p}_\theta[\beta/\lambda](x_{t-1}\,|\,x_t,c) = \frac{p_{\text{ref}}(x_{t-1}\,|\,x_t,c)^{1-\lambda}\ p_\theta^*[\beta](x_{t-1}\,|\,x_t,c)^\lambda}{\int p_{\text{ref}}(x_{t-1}'\,|\,x_t,c)^{1-\lambda}\ p_\theta^*[\beta](x_{t-1}'\,|\,x_t,c)^\lambda\,dx_{t-1}'}. \tag{5}$$

**Interpretation.** Equation 4 blends reference and aligned densities by raising each to complementary powers. Equation 5 applies an analogous idea at each denoising step, enabling sampling with the effect of alignment without retraining. The parameter $\lambda$ controls the KL regularization strength. When $\lambda = 0$, the regularization strength $\beta/\lambda$ is infinite, thus recovering the original $p_{\text{ref}}$ model (as seen in Equation 4). When $\lambda = 1$, we have $\beta/\lambda = \beta$, which recovers the aligned model $p_\theta[\beta]$. When $0 < \lambda < 1$, the new model $\hat{p}_\theta[\beta/\lambda]$ is an interpolation between the two models, which is the most stable and yields the best performance (see Figure 3) as it is a convex combination between the log densities. When $\lambda > 1$, then $\hat{p}_\theta[\beta/\lambda]$ uses a lower regularization strength than the strength with which the anchoring model $p_\theta[\beta]$ has been trained with. However, this extrapolation process is no longer a convex combination and may cause the new covariance matrix (see theorem 1) to be non-positive definite and ill-conditioned, which can lead to deterioration in performance.

**Assumptions.** For the statements used in Theorem 1, we assume that the following are true: (i) per-step posteriors are well-approximated by Gaussians (scalar or diagonal variance) and (ii) the interpolation weight $\lambda$ is in the range of $[0, 1]$ (because if $\lambda > 1$, this corresponds to extrapolation and may cause performance degradation due to absence of positive definiteness of the new covariance matrix).

## 4.2 DENOISING TIME REALIGNMENT

**Theorem 1** (Closed-form per-step denoising realignment). *Denoting $\mu_1 = \mu_\theta(x_t, t, c)$, $\mu_2 = \mu_\theta^*[\beta](x_t, t, c)$ and $\sigma_1^2 = \sigma_{t|t-1}^2 \frac{\sigma_{t-1}^2}{\sigma_t^2} I = \sigma_2^2$ for brevity, Let*

$$p_{ref}(x_{t-1}|x_t, c) = \mathcal{N}(x_{t-1}; \mu_1, \sigma_1^2 I) \qquad p_\theta^*[\beta](x_{t-1}|x_t, c) = \mathcal{N}(x_{t-1}; \mu_2, \sigma_2^2 I)$$

*Then, for any interpolation weight $\lambda \in [0, 1]$ the stepwise realigned posterior*

$$\hat{p}_\theta[\beta/\lambda](x_{t-1} \mid x_t, c) = \frac{p_{ref}(x_{t-1} \mid x_t, c)^{1-\lambda} \, p_\theta^*[\beta](x_{t-1} \mid x_t, c)^\lambda}{\int p_{ref}(x'_{t-1} \mid x_t, c)^{1-\lambda} \, p_\theta^*[\beta](x'_{t-1} \mid x_t, c)^\lambda \, dx'_{t-1}} \tag{6}$$

*is Gaussian with closed-form parameters:*

$$\Sigma_{new} = \left( \frac{1-\lambda}{\sigma_1^2} + \frac{\lambda}{\sigma_2^2} \right)^{-1} I \qquad \mu_{new} = \Sigma_{new} \left( \frac{(1-\lambda)}{\sigma_1^2} \mu_1 + \frac{\lambda}{\sigma_2^2} \mu_2 \right) \tag{7}$$

*Moreover, deterministic scheduler posterior transform (including schedulers used by DDIM/DDPM samplers) preserves the Gaussian form of $\hat{p}_\theta[\beta/\lambda]$, allowing the closed-form update above to be applied at each denoising step.*

**Proof sketch.** Note that $p_{ref}(x_{t-1}|x_t, c)^{1-\lambda} \, p_\theta[\beta](x_{t-1}|x_t, c)^\lambda \propto \exp\left( -\frac{1}{2} \left( \frac{1-\lambda}{\sigma_1^2} \|x_{t-1} - \mu_1\|^2 + \frac{\lambda}{\sigma_2^2} \|x_{t-1} - \mu_2\|^2 \right) \right)$. We then define $\Sigma_{new} = \left( \frac{1-\lambda}{\sigma_1^2} + \frac{\lambda}{\sigma_2^2} \right)^{-1} I$ and $\mu_{new} = \Sigma_{new} \left( \frac{(1-\lambda)}{\sigma_1^2} \mu_1 + \frac{\lambda}{\sigma_2^2} \mu_2 \right)$. Following this, one sees that the product can be written as an unnormalized Gaussian. Finally, using algebraic manipulation with respect to the integral, we arrive at a normalized Gaussian from which we can sample. We note that $\Sigma_{new}$ is guaranteed to be positive definite for $\lambda \in [0, 1]$ and $\sigma_1^2, \sigma_2^2 > 0$. Moreover, this same closed form update applies iteratively at each denoising step. We provide a full and detailed derivation which is available at appendix A.3.

**Corollary 1** (Positivity and scalar simplification). *If $\sigma_1^2, \sigma_2^2 > 0$ and $\lambda \in [0, 1]$, then $\sigma_{new}^2 > 0$ and the interpolated posterior is a valid Gaussian. In the isotropic (scalar) case, the $\sigma_{new}^2$ and $\mu_{new}$ are as follows*

$$\sigma_{new}^2 = \frac{\sigma_1^2 \sigma_2^2}{\sigma_2^2(1-\lambda) + \sigma_1^2 \lambda} \qquad \mu_{new} = \sigma_{new}^2 \left( \frac{(1-\lambda)}{\sigma_1^2} \mu_1 + \frac{\lambda}{\sigma_2^2} \mu_2 \right) \tag{8}$$

*which is implemented in algorithm 1.*

**Remark** As seen in eq. (7), $\lambda > 1$ forces a non convex combination, as such, since $1 - \lambda < 0$, it may cause the new covariance matrix to not be positive definite and ill-conditioned. But empirically, DeRaDiff continues to approximate a model with lesser effective regularization for moderate $\lambda > 1$ before instability occurs (see fig. 3).
**Multi-reward extension** We also prove that DeRaDiff can be extended to the very general case of multi-reward modelling (Ramé et al. (2023), Jang et al. (2023), Mitchell et al. (2023)). A full proof is given at appendix A.4.

---

$x_t, x_{t-1}, \mu_t, \mu_{t-1} \in \mathbb{R}^D$
Note that $\sigma_1^2$ need not be equal to $\sigma_2^2$–our derivation handles this more general case.

## 4.3 ALGORITHM

---

**Algorithm 1** DeRaDiff Sampling

---

**Require:** Reference model $\mathcal{E}_{\boldsymbol{\theta}_{\text{ref}}}$, Aligned model $\mathcal{E}_{\boldsymbol{\theta}_{\text{tuned}}}$, interpolation weight $\lambda \in [0, 1]$, prompt $p$, guidance scale $\gamma$, number of inference steps $N$, scheduler with timesteps $\{t_i\}_{i=0}^{N}$ and corresponding noise levels $\{\sigma_i\}_{i=0}^{N}$.

1: $c \leftarrow \text{Encode}(p)$
2: $c_{\text{null}} \leftarrow \text{Encode}(\text{``"})$           $\triangleright$ Get unconditional embedding
3: $\boldsymbol{x}_{t_N} \sim \mathcal{N}(\boldsymbol{0}, \boldsymbol{I})$       $\triangleright$ Sample initial latent from a standard Gaussian distribution
4: **for** $i = N, \ldots, 1$ **do**
5:    $t \leftarrow t_i, t_{\text{prev}} \leftarrow t_{i-1}$
6:    $\sigma_t \leftarrow \sigma_i$
7:    $\boldsymbol{\epsilon}^{\text{ref}} \leftarrow \mathcal{E}_{\boldsymbol{\theta}_{\text{ref}}}(\boldsymbol{x}_t, \sigma_t, \boldsymbol{c}_{\text{null}}) + \gamma \left( \mathcal{E}_{\boldsymbol{\theta}_{\text{ref}}}(\boldsymbol{x}_t, \sigma_t, \boldsymbol{c}) - \mathcal{E}_{\boldsymbol{\theta}_{\text{ref}}}(\boldsymbol{x}_t, \sigma_t, \boldsymbol{c}_{\text{null}}) \right)$
8:    $\boldsymbol{\epsilon}^{\text{tuned}} \leftarrow \mathcal{E}_{\boldsymbol{\theta}_{\text{tuned}}}(\boldsymbol{x}_t, \sigma_t, \boldsymbol{c}_{\text{null}}) + \gamma \left( \mathcal{E}_{\boldsymbol{\theta}_{\text{tuned}}}(\boldsymbol{x}_t, \sigma_t, \boldsymbol{c}) - \mathcal{E}_{\boldsymbol{\theta}_{\text{tuned}}}(\boldsymbol{x}_t, \sigma_t, \boldsymbol{c}_{\text{null}}) \right)$   $\triangleright$ Compute Classifier-Free Guidance predictions for both models.
9:    $\boldsymbol{\mu}_1, \sigma_1^2 \leftarrow \text{SchedulerPosterior}(\boldsymbol{x}_t, \boldsymbol{\epsilon}^{\text{ref}}, t, t_{\text{prev}})$
10:    $\boldsymbol{\mu}_2, \sigma_2^2 \leftarrow \text{SchedulerPosterior}(\boldsymbol{x}_t, \boldsymbol{\epsilon}^{\text{tuned}}, t, t_{\text{prev}})$   $\triangleright$ Calculate posterior mean $\boldsymbol{\mu}$ and variance $\sigma^2$ for the distribution at $t_{\text{prev}}$.
11:    $\sigma_{\text{new}}^2 \leftarrow \left( \frac{1-\lambda}{\sigma_1^2} + \frac{\lambda}{\sigma_2^2} \right)^{-1}$
12:    $\boldsymbol{\mu}_{\text{new}} \leftarrow \sigma_{\text{new}}^2 \left( \frac{1-\lambda}{\sigma_1^2} \boldsymbol{\mu}_1 + \frac{\lambda}{\sigma_2^2} \boldsymbol{\mu}_2 \right)$
13:    $\boldsymbol{z} \sim \mathcal{N}(\boldsymbol{0}, \boldsymbol{I})$
14:    $\boldsymbol{x}_{t_{\text{prev}}} \leftarrow \boldsymbol{\mu}_{\text{new}} + \boldsymbol{z} \cdot \sqrt{\sigma_{\text{new}}^2}$
15: **end for**
16: $\boldsymbol{I}_{\text{out}} \leftarrow \text{VAE.decode}(\boldsymbol{x}_{t_0})$
17: **return** $\boldsymbol{I}_{\text{out}}$

---

## 5 EXPERIMENTS

### 5.1 EXPERIMENTAL SETUP

Our experiments constitute the following steps:

1. **Obtain reference and realigned models**. We obtain public releases of model checkpoints (SDXL 1.0) and initialize the reference model $p_{\text{ref}}$. Then, we use an arbitrary alignment method (eg: DiffusionDPO, Wallace et al. (2023)) to align the reference model while minimizing the KL divergence where the regularization strength is $\beta$, which yields the realigned model $p_\theta[\beta]$. We also perform experiments on Stable Diffusion 1.5, which can be found in appendix A.8.

2. **Obtain outputs from Denoising Time Realignment**. For given prompts $c$, we apply algorithm 1 with varying $\lambda$ values to obtain samples from $\hat{p}_\theta[\beta/\lambda]$, allowing us to approximate various different regularization strengths without alignment from scratch.

3. **Compare denoising time realignment samples against retrained models.** We compare the downstream reward achieved by samples generated from $\hat{p}_\theta[\beta/\lambda]$ to those of $p_\theta[\beta/\lambda]$ which is a model that is aligned completely from scratch.

To comprehensively assess DeRaDiff's ability to approximate the performance of models aligned from scratch, we sample a batch of 500 prompts from a union of the Pick-a-Pic v1 and HPS datasets and test DeRaDiff's approximation capability on three metrics which cover various aspects of image generation, namely PickScore, HPS v2 and CLIP. The SDXL 1.0 model is aligned at a wide range of regularization strengths $\beta \in \{500, 1000, 2000, 5000, 8000, 10000\}$, and at a time, one aligned model at a particular $\beta$ is used as an anchor model to approximate other alignment strengths. We do the same for SD1.5, whose results are provided in detail in appendix A.8.

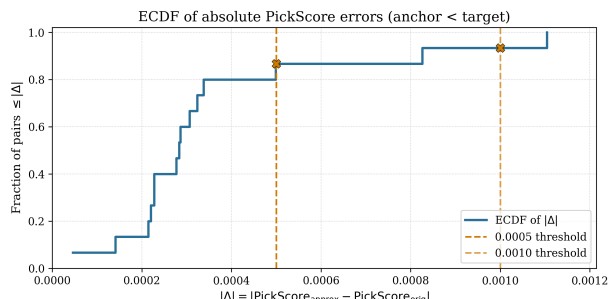

Figure 4: ECDF of absolute PickScore errors $|\Delta| = |\text{PickScore}_{\text{approx}} - \text{PickScore}_{\text{orig}}|$, when DeRaDiff is used on aligned SDXL models.

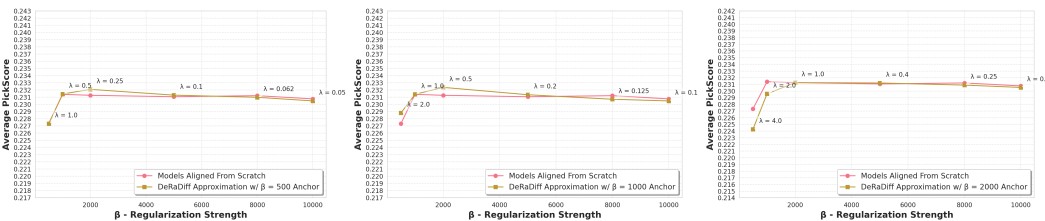

Figure 5: Line graphs for the average PickScore rewards gained by SDXL models realigned from scratch along with line graphs for the average PickScore rewards gained from DeRaDiff using anchor SDXL models with $\beta = 500$ (left plot), $\beta = 1000$ (middle plot) and $\beta = 2000$ (right plot) regularization strengths.

### 5.1.1 PICKSCORE

PickScore (Kirstain et al., 2023) is a caption-aware image reward model trained under a Bradley–Terry objective on pairwise preferences. Given a tuple $(\mathbf{c}, I_A, I_B, y)$, where $\mathbf{c}$ is the prompt, $I_A$ and $I_B$ are candidate images, and $y \in \{0, 1\}$ indicates whether $I_A$ is preferred, a CLIP-based encoder with an MLP head produces a real-valued score $s_\theta(\mathbf{c}, I)$. The induced preference probability is $\Pr(I_A \succ I_B \mid \mathbf{c}) = \sigma\big(s_\theta(\mathbf{c}, I_A) - s_\theta(\mathbf{c}, I_B)\big)$ with $\sigma(\cdot)$ the logistic sigmoid. We use PickScore as a learned reward targeting human-perceived quality under the provided caption; unless otherwise stated, higher indicates stronger preference.

As seen in fig. 4, the typical approximation error is extremely small (median $= 2.83 \times 10^{-4}, \approx 20\%$ of the PickScore std) when DeRaDiff approximates human appeal to images on aligned SDXL models. Roughly $87\%$ of approximations have errors $\leq 5 \times 10^{-4}$, so DeRaDiff produces near-identical PickScore ratings for the vast majority of cases, meaning the human appeal of images produced by DeRaDiff and models aligned entirely from scratch are near-identical.

As seen in fig. 5, DeRaDiff is able to meaningfully control the regularization strength on the fly without retraining by closely matching the SDXL models that were aligned entirely from scratch. Thus DeRaDiff enables testing of various regularization strengths without training, allowing one to search for the optimal strength, eliminating the need for expensive alignment sweeps. Moreover, using DeRaDiff, one can identify a promising range of regularization strengths and *only align at these strengths*, substantially reducing computational costs.

### 5.1.2 HPS V2

Human Preference Score v2 (HPS v2) (Wu et al., 2023b) is a caption-aware preference model trained on the Human Preference Dataset v2 (HPD v2), a large-scale corpus of pairwise judgments designed to approximate human ratings of text-to-image outputs. HPD v2 comprises on the order of $7.9 \times 10^5$ binary choices over $\sim 4.3 \times 10^5$ prompt–image pairs spanning real photographs and generations from diverse T2I models. To this end, we test how well DeRaDiff matches human-preference be-

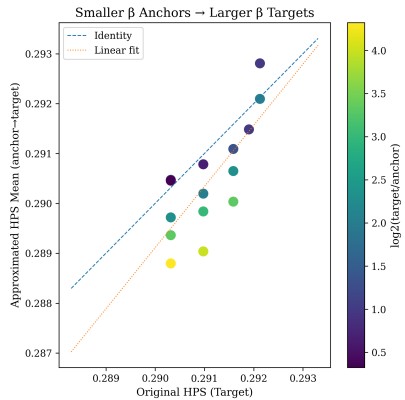 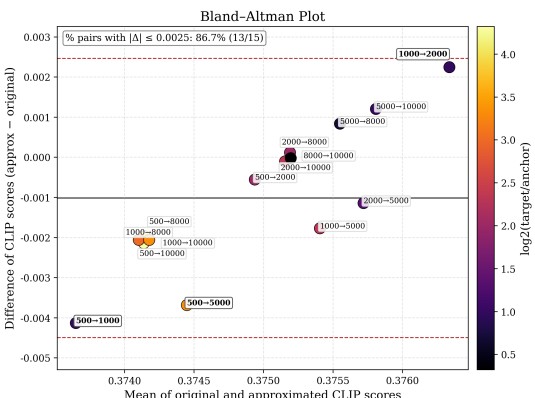

(a) Scatter plot of approximated HPS Mean vs. original HPS (target) shows DeRaDiff approximations closely match human preference scores.

(b) Bland–Altman plot of DeRaDiff approximations for mean CLIP scores showing no systematic semantic fidelity approximation bias.

Figure 6: Graphical plots of statistical analysis of DeRaDiff's approximations.

haviours of models realigned from scratch by presenting and analyzing a scatter plot. As seen in fig. 6a, each point $(x, y)$, corresponds to a a specific approximation of a $\beta_{target}$ SDXL model using a specific $\beta_{anchor}$ anchoring SDXL model using DeRaDiff where $x$ corresponds to the mean HPS score obtained by the DeRaDiff approximation of the $\beta_{target}$ model and $y$ corresponds to the mean HPS score obtained by the $\beta_{target}$ model. Moreover, we colour code each point $(x, y)$ with its respective $\log_2(\beta_{target}/\beta_{anchor})$ value with the goal of encoding the gap between the regularization strengths of $\beta_{anchor}$ and $\beta_{target}$. Here we see that points lie around the identity line and the linear fit is close to it. This indicates that DeRaDiff is able to match and recover the human-preference scores of images from models aligned entirely from scratch. Moreover, inferring from the color scale, this indicates that approximations is near-identical or even better when an anchor $\beta$ approximates a target $\beta$ that is close-by, but performance degrades smoothly with increasing anchor-to-target distance. Overall, this figure provides a faithfulness check: DeRaDiff enables low-cost, inference-time alignment that is able to preserve human preference outcomes of models aligned entirely from scratch. Detailed statistical analysis is provided in appendix A.8.

### 5.1.3 CLIP

CLIP (Hessel et al., 2021) provides a general-purpose text–image relevance score without explicit training on human preference pairs. For a caption–image pair $(p, x)$, we compute the cosine similarity of normalized embeddings, $s_{\text{CLIP}}(p, x) = \frac{\text{Enc}_{\text{text}}(p) \cdot \text{Enc}_{\text{img}}(x)}{\|\text{Enc}_{\text{text}}(p)\| \, \|\text{Enc}_{\text{img}}(x)\|}$. In our evaluations, CLIP is treated as a semantic fidelity baseline to complement preference-trained metrics (HPS v2, PickScore), helping to disentangle prompt adherence from aesthetic appeal. To demonstrate how DeRaDiff maintains semantic fidelity and that DeRaDiff has no systematic semantic approximation bias when considering the preservation of semantic-fidelity, we present a Bland-Altman comparison for DeRaDiff approximations on SDXL models in fig. 6b. In this Bland-Altman plot, for each point $(x, y)$ with label $\beta_{anchor} \to \beta_{target}$, $x$ refers to the average of (a) the CLIP score gained by the DeRaDiff approximation of a $\beta_{target}$ reference model using a $\beta_{anchor}$ SDXL model as the anchor and (b) the original CLIP score gained by the target $\beta_{target}$ SDXL model. And $y$ refers to the difference between (a) and (b), i.e. the difference between the CLIP score gained by the DeRaDiff approximation and the CLIP score gained by the target model that was aligned from scratch. We use a similar colour scheme for each point as was described in section 5.1.2. Here, fig. 6b demonstrates that DeRaDiff approximations have negligible average bias and show very small absolute differences (maximum $|\Delta| \approx 4.5 \times 10^{-3}$, 1.2% of $\mu_{orig}$, where $\mu_{orig}$ is the mean of all CLIP values generated by models aligned completely from scratch) and that the Bland-Altman mean difference is -0.001018, which is -0.273% of $\mu_{orig}$. Furthermore, the 95% limits of agreement is $[-0.004496, 0.002461]$. Further analysis is provided in appendix A.8. These results indicate that

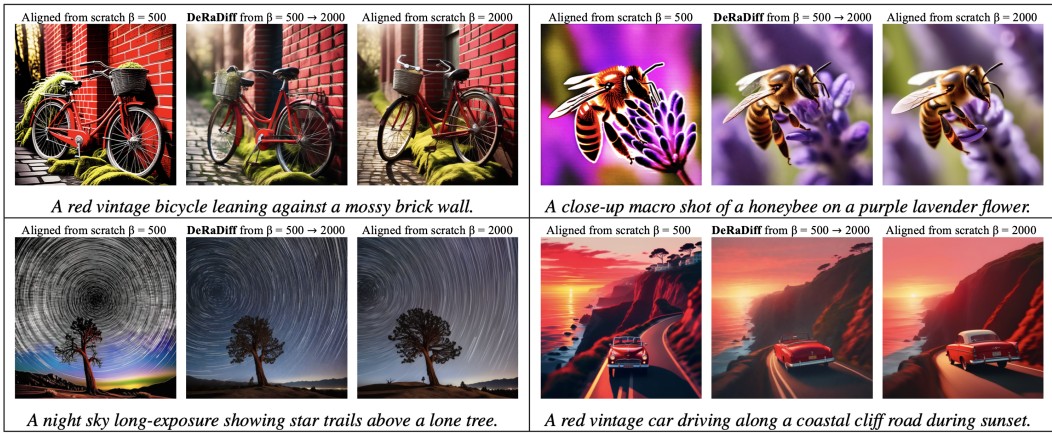

Figure 7: **DeRaDiff undoes reward hacking.** For each panel, left = image from SDXL model aligned at $\beta = 500$ (reward-hacked), center = DeRaDiff approximating an SDXL model aligned at $\beta = 2000$ using an SDXL anchor aligned at $\beta = 500$, right = reference image from an SDXL model aligned at $\beta = 2000$. The image details and style are successfully recovered by DeRaDiff.

DeRaDiff preserves prompt-to-image semantic fidelity with no systematic bias, particularly when $\lambda \in [0, 1]$. Taken together, these results show that DeRaDiff preserves prompt-to-image semantic fidelity to within measurement noise in CLIP, thus rendering DeRaDiff once more capable of tuning the regularization strength on the fly during inference accurately preserving semantics all the while obviating the need to perform multiple costly retrainings.

## 5.2 QUALITATIVE ANALYSIS

As seen in table 1, DeRaDiff is capable of producing highly accurate training-free approximations particularly in the case for $\lambda \in [0, 1]$, and is able to meaningfully control the regularization strength at inference time on the fly. We provide further detailed statistical analysis in appendix A.8. Across both SDXL and SD1.5, mean absolute errors are extremely small (all $< 0.02$ in absolute terms) and remain well below $0.5\%$ when taken with respect to the respective means. The results show that DeRaDiff reproduces the average behavior of models aligned entirely from scratch for the case $\lambda \in [0, 1]$. In table 2, we show the performance of DeRaDiff on an arbitrary anchor aligned at $\beta = 2000$. We observe that the performance of DeRaDiff is generally stronger when applied to approximating models aligned with regularization strengths that are higher than that of the anchor model. This is explained by the fact that when $0 \leq \lambda \leq 1$, DeRaDiff performs a convex combination, as seen in Equation 5. The experiments show that this interpolation is stable and is thus a reliable surrogate to approximate the performance of models aligned at such regularization strengths. When $\lambda > 1$, the combination is not convex as discussed in theorem 1. This leads to slightly less accurate approximations. Furthermore, as seen in fig. 5 and fig. 7, our experiments demonstrate that DeRaDiff can provide a reliable approximation of models aligned from scratch even when using a reward hacked model as the anchor. As reward hacked models have small $\beta$ values, we can undo the effect of reward hacking by using the reward hacked model as an anchor and utilise a small $\lambda$ value to reverse the effect of reward hacking (as seen in fig. 7). However, note that due to the stepwise denoising approximation and numerical approximation errors, one must also expect to see certain cases where the re-approximation may not be visually similar, even though the RLHF scores of the re-approximation closely match those from a model aligned completely from scratch. We further provide detailed evaluations of DeRaDiff's capability to undo reward hacking in appendix A.6.2.

## 6 COMPUTE SAVINGS

In our experimental setup detailed in appendix A.7, aligning a SDXL model at a single $\beta$ takes $\approx 336$ GPU hours, which is $\approx 52,416$ TFLOP-hours (FP16 Tensor-core equivalent) at a sustained load of $50\%$, or $\approx 1.887 \times 10^{20}$ floating point operations ($\approx 188.7$ EFLOPs). If a naive pipeline

**Table 1:** Training-free approximation errors of DeRaDiff when $\lambda \in [0, 1]$

| Model | CLIP | | HPS | | PickScore | |
|---|---|---|---|---|---|---|
| | MAE | MAE (% of $\mu$) | MAE | MAE (% of $\mu$) | MAE | MAE (% of $\mu$) |
| SDXL | 0.001 604 | 0.430 000 | 0.000 770 | 0.265 000 | 0.000 355 | 0.154 000 |
| SD1.5 | 0.001 557 | 0.448 000 | 0.001 175 | 0.425 000 | 0.000 718 | 0.332 000 |

**Notes:** MAE = mean absolute error between DeRaDiff outputs and images generated by models aligned from scratch across all regularization strength anchors. For each metric and model, $\mu$ is the evaluated mean metric value when aligned from scratch across all evaluated regularization strengths; reported percentages are MAE divided by $\mu$. A very detailed statistical analysis is provided in

| Tasks (Anchor $\beta = 2000$) | Target Model $\beta$-values | | | | | |
|---|---|---|---|---|---|---|
| | 500 | 1000 | 2000 | 5000 | 8000 | 10000 |
| **PickScore** | | | | | | |
| Actual | 0.2273 | 0.2314 | 0.2313 | 0.2311 | 0.2312 | 0.2308 |
| Approximated | 0.2243 | 0.2296 | 0.2313 | 0.2312 | 0.2309 | 0.2305 |
| Absolute Difference (%) | 1.3451 | 0.7831 | 0.0000 | 0.0611 | 0.1399 | 0.0987 |
| **HPS** | | | | | | |
| Actual | 0.2869 | 0.2919 | 0.2921 | 0.2916 | 0.2910 | 0.2903 |
| Approximated | 0.2852 | 0.2918 | 0.2921 | 0.2911 | 0.2902 | 0.2897 |
| Absolute Difference (%) | 0.5890 | 0.0299 | 0.0000 | 0.1701 | 0.2688 | 0.2061 |
| **CLIP** | | | | | | |
| Actual | 0.3628 | 0.3757 | 0.3752 | 0.3763 | 0.3751 | 0.3752 |
| Approximated | 0.3643 | 0.3738 | 0.3752 | 0.3751 | 0.3752 | 0.3751 |
| Absolute Difference (%) | 0.4022 | 0.5077 | 0.0000 | 0.3041 | 0.0310 | 0.0282 |

**Table 2: Comparison of mean rewards achieved on various metrics by using an aligned $\beta = 2000$ SDXL model as an anchor.** DeRaDiff closely matches the models that were aligned completely from scratch. In particular, when $\lambda \leq 1$, the largest absolute percentage difference for PickScore, HPS and CLIP are 0.1399%, 0.2688%, 0.3041% respectively, thus demonstrating the accuracy of DeRaDiff's approximations.

aligns a single SDXL model at $N$ regularization strengths, the costs will scale to $N \times 336$ GPU hours (or $N \times 188.7$ EFLOPs). However, on the other hand, DeRaDiff requires aligning only *once*, thus the cumulative wall-time and FLOPs are reduced by a factor of $N$. For instance, using DeRaDiff instead of naively aligning of $N = 3, 5, 10$ yields approximate GPU-hour savings of 66.7%, 80%, and 90% respectively, and EFLOP savings of $\approx 377.4$ EFLOPs (N=3), 754.8 EFLOPs (N=5), and 1,698.3 EFLOPs (N=10) respectively. Thus, by using DeRaDiff for finding the optimal range of regularization strengths in place of a naive alignment sweep at N $\beta$'s, one is capable of reducing the run-time and FLOPs by a percentage of $\approx 1 - \frac{1}{N}$, which represents a substantial saving in computational costs and run-time. However, DeRaDiff requires two forward passes at inference, but taken in totality, this overhead is still always smaller compared to full alignment sweeps. Moreover, this inference overhead can still be reduced by using prompt encoding caching or parallelized inference.

# 7  CONCLUSION

In this work, we introduced DeRaDiff, a theoretical expansion of decoding time realignment to diffusion models, a framework enabling one to modulate the regularization strength of any aligned diffusion model on the fly without any additional training. We also provided experimental evidence that DeRaDiff enables precise and meaningful control of alignment strength and accelerates RLHF-style hyperparameter exploration while preserving downstream performance in terms of RLHF scores. We also demonstrated the substantial compute savings that DeRaDiff brings about. Thus, in conclusion, DeRaDiff yields an efficient way to search for the optimal regularization strength, eliminating the need for expensive alignment sweeps.

## 8  ETHICAL STATEMENT

The authors have read the ICLR Code of Ethics and are committed to complying and upholding them. We only note two potential concerns: (1) pretrained image models and their training data might contain copyrighted content and also it may include societal biases, and (2) by lowering the computational costs involved in alignment, this can reduce the barrier to deployment and may increase the risk of misuse. The authors wish to inform that they are strictly against such misuse and encourage responsible and safe use at all times without question. Furthermore, the authors only use pretrained models and datasets that are available to the public and are committed to strictly adhering to all model and dataset license restrictions.

## 9  REPRODUCIBILITY STATEMENT

The authors make every effort to make their work fully reproducible. To this end, the authors freely share the source code required to run our experiments at `github.com/itsShahain/DeRaDiff`. We also detail our experimental setup in appendix A.7. Furthermore, we have used publicly available SDXL and SD1.5 checkpoints.

## 10  ACKNOWLEDGEMENT

Shahain Manujith would like to thank Teoh Tze Tzun for helping with proof reading, enhancing the flow of the paper with multiple additions, helping to present mathematical results concisely, conducting final experiments and for refactoring the codebase. Shahain Manujith would also like to thank Dr. Yang Zhang sincerely for helping with research ideation, overseeing the progress of the research, providing weekly mentorship and feedback for manuscript drafts and providing access to compute.

Moreover, this material is based upon work supported by the Air Force Office of Scientific Research under award number FA2386-24-1-4011, and this research is partially supported by the Singapore Ministry of Education Academic Research Fund Tier 1 (Award No: T1 251RES2509).

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

CONTENTS

# A APPENDIX

## A.1 RE-EXPRESSION OF REALIGNED MODEL

This concerns the re-expression of the realigned model in terms of a model that was aligned from scratch. Thus, this shows a way to feasibly approximate an aligned model without training from scratch. From equation 3 we see that $e^{\frac{1}{\beta}r(c,x_0)} = Z(c)p_\theta^*[\beta](x_0|\mathbf{c})/p_{\text{ref}}(x_0|c)$ and hence

$$
\begin{aligned}
p_\theta^*[\beta/\lambda](x_0|c) &= \frac{p_{\text{ref}}(x_0|c)e^{\frac{\lambda}{\beta}r(c,x_0)}}{\int p_{\text{ref}}(x_0'|c)e^{\frac{\lambda}{\beta}r(c,x_0')}\,dx_0'} = \frac{p_{\text{ref}}(x_0|c)\left[e^{\frac{1}{\beta}r(c,x_0)}\right]^\lambda}{\int p_{\text{ref}}(x_0'|c)\left[e^{\frac{1}{\beta}r(c,x_0')}\right]^\lambda\,dx_0'} \\
&= \frac{P_{\text{ref}}(x_0|c)\left[Z(c)P_\theta^*[\beta](x_0|c)/P_{\text{ref}}(x_0|c)\right]^\lambda}{\int P_{\text{ref}}(x_0'|c)\left[Z(c)P_\theta^*[\beta](x_0'|c)/P_{\text{ref}}(x_0'|c)\right]^\lambda\,dx_0'} \\
&= \frac{P_{\text{ref}}(x_0|c)\left[P_\theta^*[\beta](x_0|c)/P_{\text{ref}}(x_0|c)\right]^\lambda}{\int P_{\text{ref}}(x_0'|c)\left[P_\theta^*[\beta](x_0'|c)/P_{\text{ref}}(x_0'|c)\right]^\lambda\,dx_0'}
\end{aligned}
\tag{9}
$$

## A.2 UNIQUE GLOBAL OPTIMUM FOR THE CONTINUOUS CASE

This concerns the finding of the unique global optimum for equation 2. This proof is as seen in Rafailov et al. (2023), but where the partition function is for the continuous case. Using the definition of the KL divergence, equation 2 simplifies to:

$$
\begin{aligned}
P_\theta^* &= \max_{\rho_\theta} \mathbb{E}_{c\sim\mathcal{D}_c}\left[\mathbb{E}_{x_0\sim p_\theta(x_0|c)}\left[r(c,x_0) - \beta\log\frac{p_\theta(x_0|c)}{p_{\text{ref}}(x_0|c)}\right]\right] \\
&= \min_{\rho_\theta} \mathbb{E}_{c\sim\mathcal{D}_c}\left[\mathbb{E}_{x_0\sim p_\theta(x_0|c)}\left[\log\frac{p_\theta(x_0|c)}{p_{\text{ref}}(x_0|c)} - \frac{1}{\beta}r(c,x_0)\right]\right] \\
&= \min_{\rho_\theta} \mathbb{E}_{c\sim\mathcal{D}_c}\left[\mathbb{E}_{x_0\sim p_\theta(x_0|c)}\left[\log\frac{p_\theta(x_0|c)}{\frac{1}{Z(c)}p_{\text{ref}}(x_0|c)e^{\frac{1}{\beta}r(c,x_0)}} - \log Z(c)\right]\right]
\end{aligned}
\tag{10}
$$

Here, the partition function is:

$$
Z(c) = \int p_{\text{ref}}(x_0'|c)e^{\frac{1}{\beta}r(c,x_0')}dx_0'
\tag{11}
$$

Now, define $p^*(x_0|c) = \frac{1}{Z(c)}p_{\text{ref}}(x_0|c)e^{\frac{1}{\beta}r(c,x_0)}$ which is a valid probability distribution as $p^*(x_0|c) \geq 0$ for all $x_0$ and $\int p^*(x_0|c)dx_0 = 1$.

Then, since $Z(c)$ is not a function of $x_0$, bring the expectation inside:

$$
\begin{aligned}
P_\theta^* &= \min_{\rho_\theta} \mathbb{E}_{c\sim\mathcal{D}_c}\left[\mathbb{E}_{x_0\sim p_\theta(x_0|c)}\left[\log\frac{p_\theta(x_0|c)}{\frac{1}{Z(c)}p_{\text{ref}}(x_0|c)e^{\frac{1}{\beta}r(c,x_0)}}\right] - \log Z(c)\right] \\
&= \min_{\rho_\theta} \mathbb{E}_{c\sim\mathcal{D}_c}\left[\mathcal{D}_{KL}(p_\theta(x_0|c)||p^*(x_0|c)) - \log Z(c)\right]
\end{aligned}
\tag{12}
$$

Since the second term doesn't depend on on $p_\theta$, the minimum is achieved by the $p_\theta$ that minimizes the first term. Thus,

$$
p_\theta = p_\theta^* = \frac{1}{Z(c)}p_{\text{ref}}(x_0|c)e^{\frac{1}{\beta}r(c,x_0)}
\tag{13}
$$

More specifically,

$$p_\theta[\beta](x_0|c) = \frac{p_{\text{ref}}(x_0|c)e^{\frac{1}{\beta}r(c,x_0)}}{\int p_{\text{ref}}(x_0'|c)e^{\frac{1}{\beta}r(c,x_0')}dx_0'} \tag{14}$$

is a diffusion model that is aligned with a regularization strength $\beta(\neq 0)$.

### A.3 PROOF OF DENOISING TIME REALIGNMENT

This proof concerns the finding of a closed-form formula for

$$p_\theta^*[\beta/\lambda](x_{t-1}|x_t,c) = \frac{p_{\text{ref}}(x_{t-1}|x_t,c)^{1-\lambda}p_\theta^*[\beta](x_{t-1}|x_t,c)^\lambda}{\int p_{\text{ref}}(x_{t-1}'|x_t,c)^{1-\lambda}p_\theta^*[\beta](x_{t-1}'|x_t,c)^\lambda dx_{t-1}'}. \tag{15}$$

As noted previously, we have that

$$p_{\text{ref}}(x_{t-1}|x_t,c) = \mathcal{N}\left(x_{t-1}; \mu_\theta(x_t,t,c), \sigma_{t|t-1}^2 \frac{\sigma_{t-1}^2}{\sigma_t^2}\mathbf{I}\right)$$

and

$$p_\theta^*[\beta](x_{t-1}|x_t,c) = \mathcal{N}\left(x_{t-1}; \mu_\theta^*[\beta](x_t,t,c), \sigma_{t|t-1}^2 \frac{\sigma_{t-1}^2}{\sigma_t^2}\mathbf{I}\right).$$

For ease of notation, denote $\mu_1 = \mu_\theta(x_t,t,c)$, $\mu_2 = \mu_\theta^*[\beta](x_t,t,c)$ and $\sigma_1^2 = \sigma_{t|t-1}^2 \frac{\sigma_{t-1}^2}{\sigma_t^2} = \sigma_2^2$.

Using the closed form of the isotropic multivariate gaussian distribution we have that,

$$p_{\text{ref}}(x_{t-1}|x_t,c) = \mathcal{N}(x_{t-1}; \mu_1, \sigma_1^2\mathbf{I}) = \frac{\exp\left\{-\frac{1}{2\sigma_1^2}\|x_{t-1} - \mu_1\|^2\right\}}{(2\pi\sigma_1^2)^{D/2}}$$

and

$$p_\theta^*[\beta](x_{t-1}|x_t,c) = \mathcal{N}(x_{t-1}; \mu_2, \sigma_2^2\mathbf{I}) = \frac{\exp\left\{-\frac{1}{2\sigma_2^2}\|x_{t-1} - \mu_2\|^2\right\}}{(2\pi\sigma_2^2)^{D/2}}.$$

Define

$$\Sigma_{new} = \left(\frac{1-\lambda}{\sigma_1^2} + \frac{\lambda}{\sigma_2^2}\right)^{-1}\mathbf{I}, \tag{16}$$

$$\mu_{new} = \Sigma_{new}\left(\frac{(1-\lambda)}{\sigma_1^2}\mu_1 + \frac{\lambda}{\sigma_2^2}\mu_2\right). \tag{17}$$

Now considering the numerator of equation 15 , we obtain

$$p_{\text{ref}}(x_{t-1}|x_t,c)^{1-\lambda}p_\theta^*[\beta](x_{t-1}|x_t,c)^\lambda = \frac{\exp\left\{-\frac{\alpha}{2}\right\}}{(2\pi\sigma_1^2)^{(1-\lambda)D/2}(2\pi\sigma_2^2)^{\lambda D/2}} \tag{18}$$

where through the application of equation 16 and equation 17,

$$\begin{aligned}
\alpha &= \frac{(1-\lambda)}{\sigma_1^2}\|x_{t-1} - \mu_1\|^2 + \frac{\lambda}{\sigma_2^2}\|x_{t-1} - \mu_2\|^2 \\
&= \left(\frac{1-\lambda}{\sigma_1^2} + \frac{\lambda}{\sigma_2^2}\right)\|x_{t-1}\|^2 - 2\left(\frac{(1-\lambda)}{\sigma_1^2}\mu_1 + \frac{\lambda}{\sigma_2^2}\mu_2\right)\cdot x_{t-1} + \left(\frac{(1-\lambda)}{\sigma_1^2}\|\mu_1\|^2 + \frac{\lambda}{\sigma_2^2}\|\mu_2\|^2\right) \\
&= (x_{t-1} - \mu_{new})^T\Sigma_{new}^{-1}(x_{t-1} - \mu_{new}) - \mu_{new}^T\Sigma_{new}^{-1}\mu_{new} + \left(\frac{(1-\lambda)}{\sigma_1^2}\|\mu_1\|^2 + \frac{\lambda}{\sigma_2^2}\|\mu_2\|^2\right).
\end{aligned} \tag{19}$$

---

$x_t, x_{t-1}, \mu_t, \mu_{t-1} \in \mathbb{R}^D$

Note that $\sigma_1^2$ need not be equal to $\sigma_2^2$–our derivation handles this more general case.

Recalling equation 18 we now see that,

$$p_{\text{ref}}(x_{t-1}|x_t,c)^{1-\lambda}p_\theta^*[\beta](x_{t-1}|x_t,c)^\lambda = \frac{\exp\left\{-\frac{1}{2}\alpha\right\}}{(2\pi\sigma_1^2)^{(1-\lambda)D/2}(2\pi\sigma_2^2)^{\lambda D/2}}$$

$$= \varphi \exp\left\{-\frac{1}{2}(x_{t-1}-\mu_{new})^T\Sigma_{new}^{-1}(x_{t-1}-\mu_{new})\right\},$$

(20)

where

$$\varphi = \frac{\exp\left\{-\frac{1}{2}\left[\mu_{new}^T\Sigma_{new}^{-1}\mu_{new} - \left(\frac{(1-\lambda)}{\sigma_1^2}\|\mu_1\|^2 + \frac{\lambda}{\sigma_2^2}\|\mu_2\|^2\right)\right]\right\}}{(2\pi\sigma_1^2)^{(1-\lambda)D/2}(2\pi\sigma_2^2)^{\lambda D/2}}.$$

(21)

Note that expression $\varphi$ is a constant with respect to $x_{t-1}$. Similarly, we can also rewrite the denominator of equation 15 in the same way to arrive at

$$p_\theta^*[\beta/\lambda](x_{t-1}|x_t,c) = \frac{\varphi \cdot \exp\left\{-\frac{1}{2}(x_{t-1}-\mu_{new})^T\Sigma_{new}^{-1}(x_{t-1}-\mu_{new})\right\}}{\int \varphi \cdot \exp\left\{-\frac{1}{2}(x_{t-1}'-\mu_{new})^T\Sigma_{new}^{-1}(x_{t-1}'-\mu_{new})\right\} dx_{t-1}'}$$

$$= \frac{1}{(2\pi)^{D/2}|\Sigma_{new}|^{1/2}}\exp\left\{-\frac{1}{2}(x_{t-1}-\mu_{new})^T\Sigma_{new}^{-1}(x_{t-1}-\mu_{new})\right\}.$$

(22)

And thus we see that

$$\boxed{p_\theta^*[\beta/\lambda](x_{t-1}|x_t,c) = \mathcal{N}(x_{t-1};\mu_{new},\Sigma_{new})}$$

(23)

### A.4 Proof of Denoising Time Realignment when considering a Linear Combination of Multiple Rewards

We also consider the natural extension of decoding time realignment to DeRaDiff in the case of multi-reward RLHF as proposed by Ramé et al. (2023), Jang et al. (2023), Mitchell et al. (2023). Multi reward methods combine multiple models aligned independently using different rewards. Thus, consider the case of a linear combination of rewards $r_{\vec{\lambda}}$ defined by

$$r_{\vec{\lambda}}(c,x_0) = \sum_{i=1}^K \lambda_i * r_i(c,x_0),$$

(24)

where we have $K$ reward functions and where $\vec{\lambda} = (\lambda_1,...,\lambda_K) \in \mathbb{R}^K$. Then, considering the aligned model, $p_\theta^*[\beta,\vec{\lambda}]$ under $\vec{\lambda}$,

$$p_\theta^*[\beta,\vec{\lambda}](x_0|c) = \frac{p_{\text{ref}}(x_0|c)\exp\left\{\frac{1}{\beta}r_{\vec{\lambda}}(c,x_0)\right\}}{\int p_{\text{ref}}(x_0'|c)\exp\left\{\frac{1}{\beta}r_{\vec{\lambda}}(c,x_0')\right\} dx_0'}$$

$$= \frac{p_{\text{ref}}(x_0|c)\exp\left\{\frac{1}{\beta}\sum_{i=1}^K \lambda_i r_i(c,x_0)\right\}}{\int p_{\text{ref}}(x_0'|c)\exp\left\{\frac{1}{\beta}\sum_{i=1}^K \lambda_i r_i(c,x_0')\right\} dx_0'}.$$

(25)

Now, denoting $p_{i,\theta}^*[\beta](x_0|c)$ as the model obtained by aligned a reference model entirely from scratch using the $i^{\text{th}}$ reward, we have (as before) that,

$$\exp\left\{\frac{1}{\beta}r_i(c,x_0)\right\} = Z(c)p_{i,\theta}^*[\beta](x_0|c)/p_{\text{ref}}(x_0|c).$$

(26)

Then, we note that,

$$\exp\left\{\sum_{i=1}^k \lambda_i \frac{r_i(c,x_0)}{\beta}\right\} = \prod_{i=1}^k \exp\left\{\lambda_i \frac{r_i(c,x_0)}{\beta}\right\}$$

$$= \prod_{i=1}^k \left(Z(c)\frac{p_{i,\theta}^*[\beta](x_0\mid c)}{p_{\text{ref}}(x_0\mid c)}\right)^{\lambda_i}.$$

(27)

In a similar fashion,

$$\exp\left\{\sum_{i=1}^{k} \lambda_i \frac{r_i(c, x_0')}{\beta}\right\} = \prod_{i=1}^{k}\left(Z(c)\frac{p_{i,\theta}^*[\beta](x_0' \mid c)}{p_{\text{ref}}(x_0' \mid c)}\right)^{\lambda_i}. \tag{28}$$

Finally, we have that

$$p_\theta^*[\beta, \vec{\lambda}](x_0|c) = \frac{p_{\text{ref}}(x_0|c)\prod_{i=1}^{k}\left(Z(c)\frac{p_{i,\theta}^*[\beta](x_0|c)}{p_{\text{ref}}(x_0|c)}\right)^{\lambda_i}}{\int p_{\text{ref}}(x_0'|c)\prod_{i=1}^{k}\left(Z(c)\frac{p_{i,\theta}^*[\beta](x_0'|c)}{p_{\text{ref}}(x_0'|c)}\right)^{\lambda_i} dx_0'}. \tag{29}$$

Then, letting $\lambda_s = \sum_{i=1}^{K} \lambda_i$, we have that,

$$p_\theta^*[\beta, \vec{\lambda}](x_0|c) = \frac{p_{\text{ref}}(x_0|c)^{1-\lambda_s}\prod_{i=1}^{k}\left(Z(c)p_{i,\theta}^*[\beta](x_0 \mid c)\right)^{\lambda_i}}{\int p_{\text{ref}}(x_0'|c)^{1-\lambda_s}\prod_{i=1}^{k}\left(Z(c)p_{i,\theta}^*[\beta](x_0' \mid c)\right)^{\lambda_i} d\mathbf{x}_0'} \tag{30}$$

$$= \frac{p_{\text{ref}}(x_0|c)^{1-\lambda_s}\prod_{i=1}^{k}\left(p_{i,\theta}^*[\beta](x_0 \mid c)\right)^{\lambda_i}}{\int p_{\text{ref}}(x_0'|c)^{1-\lambda_s}\prod_{i=1}^{k}\left(p_{i,\theta}^*[\beta](x_0' \mid c)\right)^{\lambda_i} dx_0'}. \tag{31}$$

In a similar fashion, due to the intractability of eq. (32), consider the stepwise approximation:

$$p_\theta^*[\beta, \vec{\lambda}](x_{t-1}|x_t, c) = \frac{p_{\text{ref}}(x_{t-1}|x_t, c)^{1-\lambda_s}\prod_{i=1}^{k}\left(p_{i,\theta}^*[\beta](x_{t-1}|x_t, c)\right)^{\lambda_i}}{\int p_{\text{ref}}(x_{t-1}|x_t, c)^{1-\lambda_s}\prod_{i=1}^{k}\left(p_{i,\theta}^*[\beta](x_{t-1}|x_t, c)\right)^{\lambda_i} dx_0'}. \tag{32}$$

Now, the DeRaDiff proof follows almost immediately noting that

$$p_{i,\theta}^*[\beta](x_{t-1}|x_t, c) = \mathcal{N}(x_{t-1}; \mu_i, \sigma_i^2 \mathbf{I}) = \frac{1}{(2\pi\sigma_i^2)^{D/2}}\exp\left\{-\frac{1}{2\sigma_i^2}\|x_{t-1} - \mu_i\|^2\right\}.$$

Now considering the numerator of equation 32, we have

$$p_{\text{ref}}(x_{t-1}|x_t, c)^{1-\lambda_s}\prod_{i=1}^{k}\left(p_{i,\theta}^*[\beta](x_{t-1}|x_t, c)\right)^{\lambda_i} = \frac{\exp\left\{-\sum_{i=1}^{K}\frac{\lambda_i}{2\sigma_i^2}\|x_{t-1} - \mu_i\|^2\right\}}{\prod_{i=1}^{K}(2\pi\sigma_i^2)^{D/2}}. \tag{33}$$

Define

$$\Sigma_{\text{new}} = \left(\sum_{i=1}^{K}\frac{\lambda_i}{\sigma_i^2}\right)^{-1}\mathbf{I}, \tag{34}$$

$$\mu_{\text{new}} = \Sigma_{\text{new}}\left(\sum_{i=1}^{K}\frac{\lambda_i}{\sigma_i^2}\mu_i\right). \tag{35}$$

To simplify the exponent in the numerator of eq. (33), by defining $\alpha = \sum_{i=1}^{K}\frac{\lambda_i}{\sigma_i^2}\|x_{t-1} - \mu_i\|^2$ and applying equation 34 and equation 35, we have

$$\alpha = \sum_{i=1}^{K}\frac{\lambda_i}{\sigma_i^2}\|x_{t-1} - \mu_i\|^2$$

$$= \left(\sum_{i=1}^{K}\frac{\lambda_i}{\sigma_1^2}\right)\|x_{t-1}\|^2 - 2\left(\sum_{i=1}^{K}\frac{\lambda_i}{\sigma_i^2}\mu_i\right)\cdot x_{t-1} + \left(\sum_{i=1}^{K}\frac{\lambda_i}{\sigma_i^2}\|\mu_i\|^2\right)$$

$$= (x_{t-1} - \mu_{new})^T\Sigma_{new}^{-1}(x_{t-1} - \mu_{new}) - \mu_{new}^T\Sigma_{new}^{-1}\mu_{new} + \left(\sum_{i=1}^{K}\frac{\lambda_i}{\sigma_i^2}\|\mu_i\|^2\right). \tag{36}$$

Substituting the result above into equation 33, we yield

$$p_{\text{ref}}(x_{t-1}|x_t,c)^{1-\lambda_s} \prod_{i=1}^{k} \left( p_{i,\theta}^*[\beta](x_{t-1}|x_t,c) \right)^{\lambda_i}$$

$$= \frac{\exp\left\{-\frac{1}{2}\alpha\right\}}{\prod_{i=1}^{K}(2\pi\sigma_i^2)^{D/2}}$$

$$= \varphi \cdot \exp\left\{-\frac{1}{2}(x_{t-1}-\mu_{new})^T\Sigma_{new}^{-1}(x_{t-1}-\mu_{new})\right\}, \tag{37}$$

where

$$\varphi = \frac{\exp\left\{-\frac{1}{2}\left[\mu_{new}^T\Sigma_{new}^{-1}\mu_{new} - \left(\sum_{i=1}^{K}\frac{\lambda_i}{\sigma_i^2}\|\mu_i\|^2\right)\right]\right\}}{\prod_{i=1}^{K}(2\pi\sigma_i^2)^{D/2}}.$$

Finally, we have that

$$p_\theta^*[\beta,\vec{\lambda}](x_{t-1}|x_t,c) = \frac{\varphi \cdot \exp\left\{-\frac{1}{2}(x_{t-1}-\mu_{new})^T\Sigma_{new}^{-1}(x_{t-1}-\mu_{new})\right\}}{\int \varphi \cdot \exp\left\{-\frac{1}{2}(x'_{t-1}-\mu_{new})^T\Sigma_{new}^{-1}(x'_{t-1}-\mu_{new})\right\}dx'_{t-1}}$$

$$= \frac{1}{(2\pi)^{D/2}|\Sigma_{new}|^{1/2}}\exp\left\{-\frac{1}{2}(x_{t-1}-\mu_{new})^T\Sigma_{new}^{-1}(x_{t-1}-\mu_{new})\right\}. \tag{38}$$

which is the probability density of a Gaussian.

### A.5 An End-to-End process of finding the globally optimal $\lambda^*$

One can even extend DeRaDiff by employing Bayesian optimization to find the globally optimum $\lambda^*$ (and thus, the best regularization strength) that gives rise to the best downstream rewards. Here, we constrain $\lambda \in [0,1]$. Here, we outline the major ideas required to implement this.

We denote $p_\lambda(x)$ as the generative distribution arising from using denoising time parameter $\lambda$. Given a reward function (eg.the downstream Pickscore by Kirstain et al. (2023)), our goal is:

$$\lambda^* = \arg\max_{\lambda\in[0,1]} J(\lambda), \quad \text{where } J(\lambda) = \mathbb{E}_{\mathbf{x}\sim p_\lambda}[R(\mathbf{x})]. \tag{39}$$

Because $J(\lambda)$ is expensive to evaluate (since this requires running the model on a large batch of images and scoring using a reward function), we treat it as a black-box function and use gaussian process optimization to find $\lambda^*$ in as few evaluations as possible.

#### A.5.1 Gaussian Process Surrogate

We model the unkown objective $f(\lambda) \approx J(\lambda)$ via a gaussian process prior:

$$f(\lambda) \sim \mathcal{GP}(m(\lambda), k(\lambda,\lambda')). \tag{40}$$

For simplicity, we let $m(\lambda) = 0$. Next, we use the RBF kernel $k(\lambda,\lambda') = \sigma_f^2 \exp\left(-\frac{(\lambda-\lambda')^2}{2\ell^2}\right)$ with $l$ being the length-scale parameter and $\sigma_f^2$ being the signal variance. After $n-$ many evaluations at points $\{\lambda_i\}_{i=1}^n$ yielding noisy estimates $\hat{R}_i \approx J(\lambda_i)$, conditioning yields the exact posterior

$$\mu_n(\lambda) = k(\lambda,\boldsymbol{\lambda})[\mathbf{K}+\sigma_n^2\mathbf{I}]^{-1}\hat{\mathbf{R}} \tag{41}$$

$$\sigma_n^2(\lambda) = k(\lambda,\lambda) - k(\lambda,\boldsymbol{\lambda})[\mathbf{K}+\sigma_n^2\mathbf{I}]^{-1}k(\boldsymbol{\lambda},\lambda), \tag{42}$$

where $\boldsymbol{\lambda} = [\lambda_1,\ldots,\lambda_n]$, $\hat{\mathbf{R}} = [\hat{R}_1,\ldots,\hat{R}_n]^\top$, and $K_{ij} = k(\lambda_i,\lambda_j)$.

### A.5.2 ACQUISITION FUNCTION

To decide on which lambda value to evaluate next, $\lambda_{n+1}$, we maximize an acquisition function $a(\lambda)$ that balances exploration of the search space (high $\sigma_n$) and exploitation (high $\mu_n$):

1. Expected Improvement (EI):
   Let $f_n^+ = \max_{j \leq n} \hat{R}_j$. Then

   $$\text{EI}(\lambda) = \mathbb{E}_{f \sim \mathcal{N}(\mu_n, \sigma_n^2)}[\max\{f - f_n^+, 0\}] = (\mu_n - f_n^+)\Phi(z) + \sigma_n \phi(z), \quad (43)$$

   where $z = (\mu_n - f_n^+)/\sigma_n$, and $\Phi, \phi$ are the standard Normal CDF/PDF.

2. Upper Confidence Bound (UCB):

   $$\text{UCB}(\lambda) = \mu_n(\lambda) + \beta_n \sigma_n(\lambda),$$

   with $\beta_n$ chosen (e.g. $\beta_n = \sqrt{2\log(n^2\pi^2/6\delta)}$) to guarantee sublinear regret.

We demonstrate a run using SDXL aligned at $\beta = 500$ below. We leave this as an interesting avenue to work on in the future.

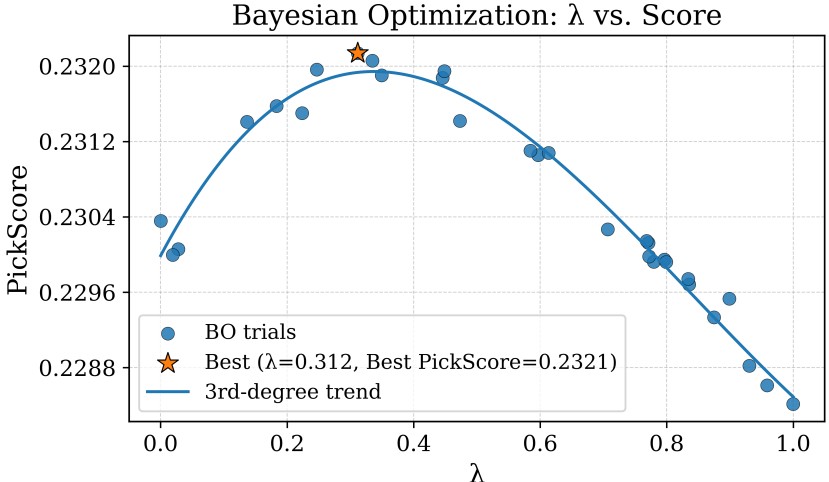

Figure 8: DeRaDiff + Bayesian optimization used to find the optimal regularization strength using an SDXL anchor model aligned at $\beta = 500$

---

**Algorithm 2** 1D Bayesian Optimization for Global $\lambda$ Selection

---

**Require:** Domain $\Lambda = [0, 1]$, budget $T$, initial design size $n_0$, reward evaluator $R(\cdot)$
**Ensure:** Best weight $\lambda^*$ and estimate $J(\lambda^*)$
 1: **Initial Design:**
 2: Sample $\{\lambda_i\}_{i=1}^{n_0} \sim \mathrm{Uniform}(\Lambda)$
 3: **for** $i = 1 \ldots n_0$ **do**
 4:      Generate batch $\{x_{i,j}\}$ from $p_{\lambda_i}$
 5:      Compute $\hat{R}_i = \frac{1}{|\{x_{i,j}\}|} \sum_j R(x_{i,j})$
 6: **end for**
 7: Fit GP surrogate on $\{(\lambda_i, \hat{R}_i)\}_{i=1}^{n_0}$
 8: **for** $t = n_0 + 1 \ldots T$ **do**
 9:      Compute posterior mean $\mu_{t-1}(\lambda)$ and variance $\sigma_{t-1}^2(\lambda)$
10:      Select next point via 1-D line search

$$\lambda_t = \arg\max_{\lambda \in \Lambda} \underbrace{\mathrm{EI}(\lambda \mid \mu_{t-1}, \sigma_{t-1})}_{\text{or UCB}}$$

11:      Generate batch from $p_{\lambda_t}$, compute $\hat{R}_t$
12:      Append $(\lambda_t, \hat{R}_t)$ to data and update GP
13: **end for**
14: **return** $\lambda^* = \arg\max_{i \leq T} \hat{R}_i$

---

### A.6    ADDITIONAL EXPERIMENTS

#### A.6.1    A FINE GRAINED EXAMINATION

In this section, we train further $\beta$ values in the interesting region of $100 \leq \beta \leq 1500$ where the human appeal rises fastest. Formally, we sample the following $\beta$ values: $250, 500, 750, 1000, 1250, 1500, 2000$ and evaluate the performance of DeRaDiff on PickScore using the experimental method detailed in section 5.1:

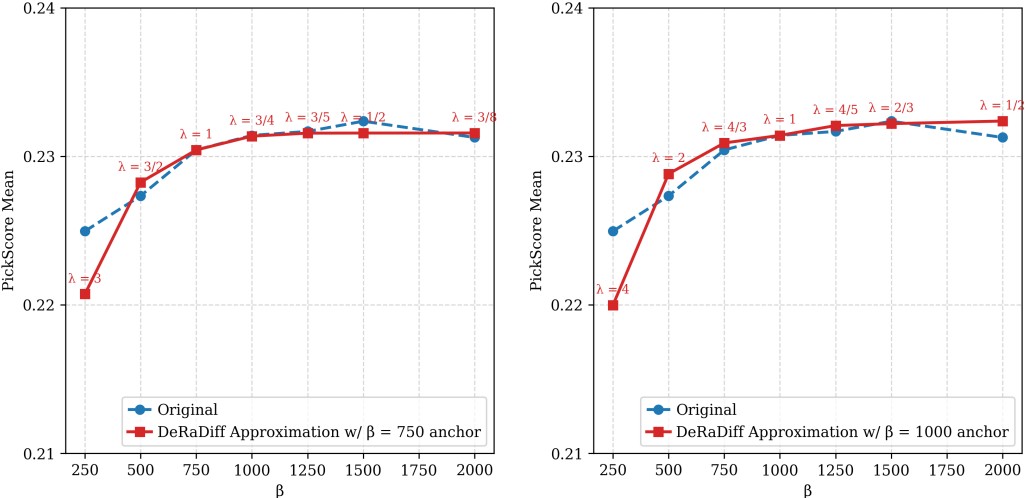

Figure 9: Line graphs for the average PickScore rewards gained by SDXL models realigned from scratch and also from DeRaDiff using anchor SDXL models with $\beta = 750$ (left plot) and $\beta = 1000$ (right plot) regularization strengths.

We see that this demonstrates a consistent increase in the perceived human appeal (measured via PickScore) for both DeRaDiff approximations and as well as for models that were realigned entirely scratch showing that DeRaDiff can faithfully re-approximate models realigned entirely from scratch.

### A.6.2 UNDOING REWARD HACKING

To demonstrate the capability of undoing reward hacking we use three reward hacked models. Namely, we use the SDXL models aligned at $\beta = 250$ (severely reward hacked), $\beta = 500$ (moderately reward hacked), $\beta = 750$ (mildly reward hacked). We use a $\beta = 2000$ model as our reference model that is healthy (i.e. not reward hacked). Since reward hacking manifests itself as drastic distributional changes in contrast, vibrancy, colour, we use the Fréchet Inception Distance to measure the degree of reward hacking and the extent to which DeRaDiff undoes reward hacking. To this end, we sample a batch of 1000 prompts from a combined dataset of HPSv2 and Partiprompts. Next, we measure the FID score between the outputs of the reward hacked models on these 1000 prompts giving rise to an average FID score for each reward-hacked model (in comparison to the healthy $\beta = 2000$ model). Next, for each reward hacked model, we use it as an anchor to re-approximate the $\beta = 2000$ model and measure the average FID score. For each model, the difference in its respective FID scores will measure the extent of undoing reward hacking.

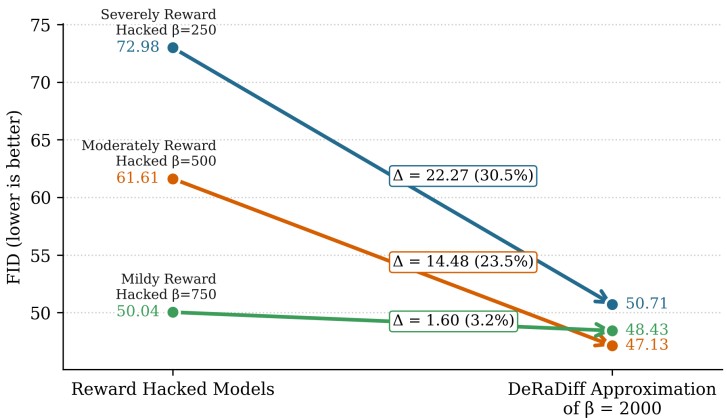

Figure 10: DeRaDiff successfully undoes reward hacking

**Table 3:** DeRaDiff undoes reward-hacking measured by FID (lower is better).

| Model (reward-hacked) | FID (reward hacked model against $\beta = 2000$) | FID (DeRaDiff approx. against $\beta = 2000$) | $\Delta$ (abs) | $\Delta$ (%) |
|---|---|---|---|---|
| Severely reward-hacked ($\beta = 250$) | 72.98 | 50.71 | 22.27 | 30.50 |
| Moderately reward-hacked ($\beta = 500$) | 61.61 | 47.13 | 14.48 | 23.50 |
| Mildly reward-hacked ($\beta = 750$) | 50.04 | 48.43 | 1.61 | 3.20 |

Thus we see that DeRaDiff is capable of undoing reward hacking and this effect of reward hacking is much more pronounced in models that are severely reward hacked. In fig. 11 we also provide more qualitative examples of how DeRaDiff can even undo extreme reward hacking.

### A.7 DETAILED EXPERIMENTAL SETUP

For our experiments, we obtain public releases of Stable Diffusion 1.5 (SD1.5) from `runwayml/stable-diffusion-v1-5` and Stable Diffusion XL 1.0 (SDXL) from `stabilityai/stable-diffusion-xl-base-1.0`. We align them using DiffusionDPO (Wallace et al., 2023) on the Pick-a-Pic v1 dataset (Kirstain et al., 2023). Each alignment is performed on 2x NVIDIA A-100 80GB GPUs. The exact hyperparameters we used are as follows:

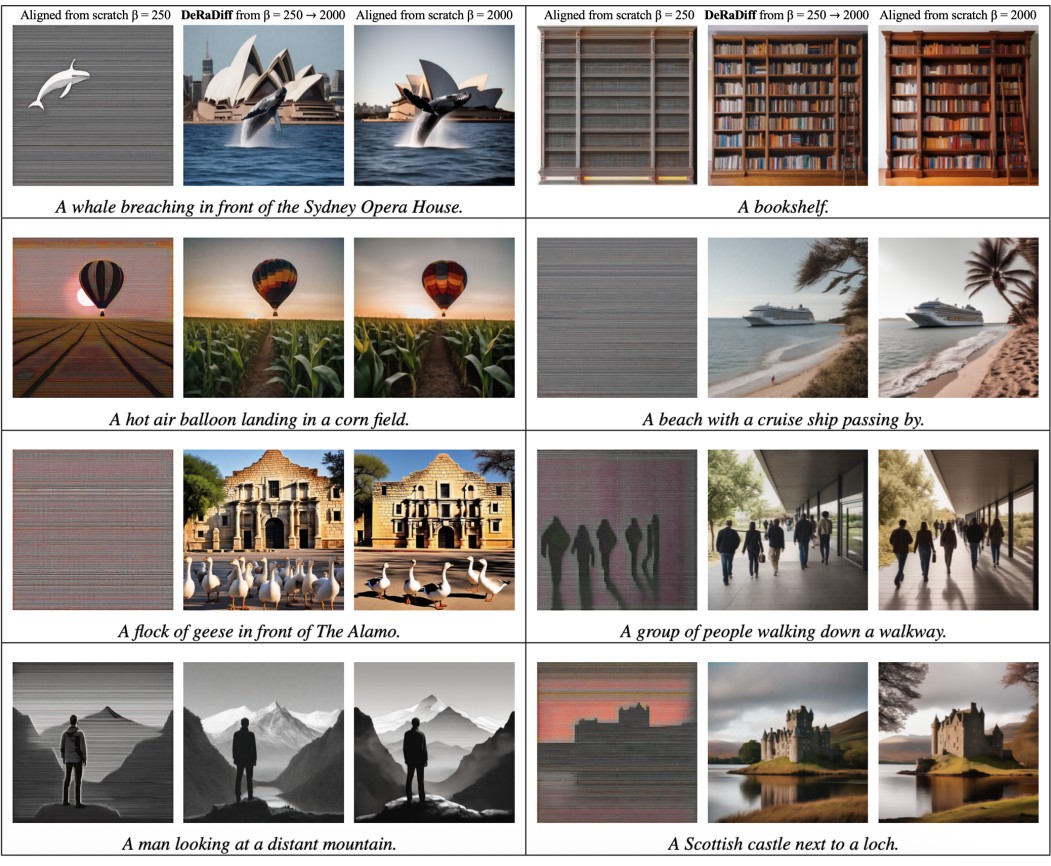

Figure 11: **DeRaDiff undoes reward hacking** For each panel, left = image from SDXL model aligned at $\beta = 250$ (severely reward-hacked), center = DeRaDiff approximating an SDXL model aligned at $\beta = 2000$ using the reward hacked SDXL anchor aligned at $\beta = 250$, right = reference image from an SDXL model aligned at $\beta = 2000$. The image details and style are successfully recovered by DeRaDiff.

**Table 4:** Aligning Hyperparameters for SDXL and SD1.5)

| Parameter | SDXL | SD1.5 |
|---|---|---|
| Pretrained VAE | madebyollin/sdxl-vae-fp16-fix | — |
| GPUs | 2 | 1 |
| Per-device batch size | 1 | 1 |
| Gradient accumulation | 64 | 64 |
| **Effective global batch size**[a] | **128** | **64** |
| Dataloader workers | 16 | 16 |
| Max train steps | 30000 | 2000 |
| LR scheduler | constant_with_warmup | constant_with_warmup |
| LR warmup steps | 200 | 500 |
| Learning rate | $1 \times 10^{-8}$ | $1 \times 10^{-8}$ |

[a] Effective global batch size = $N_{GPUs} \times$ train_batch_size $\times$ gradient_accumulation_steps. Thus SDXL: $2 \times 1 \times 64 = 128$, SD1.5: $1 \times 1 \times 64 = 64$.

Moreover, we use the Euler Ancestral Discrete scheduler for both SDXL and SD1.5. We use 50 denoising steps for both SDXL and SD1.5. Moreover, we use a guidance scale of 5 for SDXL and 7.5 for SD1.5

## A.8 Metrics used for Detailed Statistical Analysis

In this section, we provide all numerical values of original and approximated metrics and we also give a detailed statistical analysis for each.

| Metric | Formula | Meaning / interpretation |
|---|---|---|
| Mean absolute error (MAE) | $\text{MAE} = \frac{1}{n}\sum_{i=1}^{n}\lvert y_i - x_i\rvert$ | Average magnitude of the errors (unsigned). Provides a simple, easy-to-interpret measure of typical error size. |
| MAE (bootstrap mean) | $\overline{\text{MAE}}_{\text{boot}} = \frac{1}{B}\sum_{b=1}^{B}\text{MAE}^{(b)}$ | Average of MAE computed across bootstrap resamples; indicates stability of the MAE estimate under resampling. |
| MAE 95% CI (bootstrap) | $\left[\text{MAE}_{2.5\%},\ \text{MAE}_{97.5\%}\right]$ | 2.5th and 97.5th percentiles of the bootstrap MAE distribution; a 95% interval expressing sampling uncertainty. |
| Root mean squared error (RMSE) | $\text{RMSE} = \sqrt{\frac{1}{n}\sum_{i=1}^{n}(y_i - x_i)^2}$ | Similr to the MAE but squares errors first, so it penalizes larger errors more strongly (sensitive to outliers). |
| Median absolute error | $\text{MedAbs} = \text{median}\left(\lvert y_i - x_i\rvert\right)$ | The median of absolute errors; a robust measure of a "typical" error that is less sensitive to outliers. |
| Bland–Altman mean difference (bias) | $\bar{d} = \frac{1}{n}\sum_{i=1}^{n}d_i,\quad d_i = y_i - x_i$ | Mean signed difference between prediction and truth. |
| Bland–Altman SD of differences | $s_d = \sqrt{\frac{1}{n-1}\sum_{i=1}^{n}(d_i - \bar{d})^2}$ | Sample standard deviation of the differences; this simply quantifies the variability of the errors. |
| Limits of agreement (LoA) | $\text{LoA} = \bar{d} \pm 1.96\,s_d$ | Approximate interval containing $\sim$95% of individual differences (under approximate normality of differences). Useful to see practical worst-case error bounds. |
| Relative to mean (original) (%) | $\text{Rel}(M) = 100 \times \frac{M}{\bar{x}_{\text{orig}}},\quad \bar{x}_{\text{orig}} = \frac{1}{m}\sum_{j=1}^{m}x_j^{\text{orig}}$ | Expresses a metric $M$ (on the original scale) as a percentage of the mean of the original signal. This will help give n intuitive context for magnitude. |

**Table 5:** Definitions and their interpretations. Here $x_i$ denotes the true/original value, $y_i$ the approximated value, $n$ the number of pairs, $B$ bootstrap resamples, and $m$ the number of original observations whose mean is used for scaling.

## A.9 STATISTICAL ANALYSIS OF DERADIFF'S PERFORMANCE ON CLIP

### A.9.1 SDXL

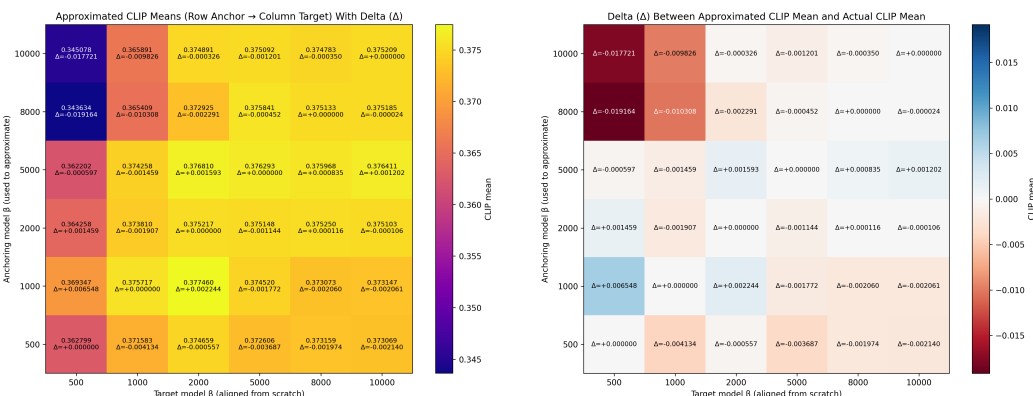

Figure 12: Approximated CLIP Means with all (Row Anchor $\beta \rightarrow$ Column Target $\beta$) with Delta ($\Delta$)

Figure 13: Delta ($\Delta$) Between Approximated CLIP Mean and Actual PickScore Mean

**Table 6:** Summary error metrics for DeRaDiff approximations for CLIP scores when $\lambda \in [0, 1]$

| Metric | Value | Relative to mean(original) (%) |
|---|---|---|
| Mean absolute error (MAE) | 0.001 604 | 0.430 000 % |
| MAE (bootstrap mean) | 0.001 608 | 0.431 000 % |
| MAE 95% CI (bootstrap) | 0.001 038 − 0.002 226 | 0.278 000 − 0.596 000 % |
| Root mean squared error (RMSE) | 0.001 994 | 0.534 000 % |
| Median absolute error | 0.001 772 | 0.475 000 % |
| Bland–Altman mean difference (mean of $y - x$) | −0.001 018 | −0.273 000 % |
| Bland–Altman SD of differences | 0.001 775 | 0.475 000 % |
| Limits of agreement (mean $\pm$ 1.96 SD) | −0.004 496 − 0.002 461 | −1.204 000 − 0.659 000 % |

Notes: *Value* columns report absolute errors on the same scale as the original data. *Relative* column uses mean(original) = 0.373 395. Limits of agreement are computed as mean difference $\pm 1.96 \times$ SD.

**Table 7:** Summary error metrics for DeRaDiff approximations for CLIP scores when $\lambda > 1$

| Metric | Value | Relative to mean(original) (%) |
|---|---|---|
| Mean absolute error (MAE) | 0.005 014 | 1.342 690 % |
| MAE (bootstrap mean) | 0.005 069 | 1.357 464 % |
| MAE 95% CI (bootstrap) | 0.002 202 − 0.008 347 | 0.589 843 − 2.235 382 % |
| Root mean squared error (RMSE) | 0.007 937 | 2.125 601 % |
| Median absolute error | 0.001 593 | 0.426 633 % |
| Bland–Altman mean difference (mean of $y - x$) | −0.003 733 | −0.999 878 % |
| Bland–Altman SD of differences | 0.007 250 | 1.941 581 % |
| Limits of agreement (mean $\pm$ 1.96 SD) | −0.017 943 − 0.010 476 | −4.805 378 − 2.805 622 % |

Notes: *Value* columns report absolute errors on the same scale as the original data. *Relative* column uses mean(original) = 0.373 395. Limits of agreement are computed as mean difference $\pm 1.96 \times$ SD.

## A.9.2 SD1.5

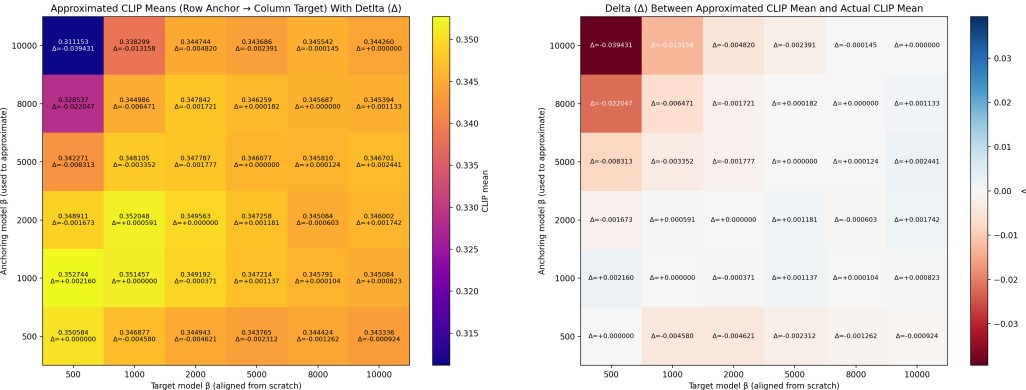

Figure 14: Approximated CLIP Means with all (Row Anchor $\beta \rightarrow$ Column Target $\beta$) with Delta $(\Delta)$

Figure 15: Delta $(\Delta)$ Between Approximated CLIP Mean and Actual CLIP Mean

**Table 8:** Summary error metrics for DeRaDiff approximations for CLIP scores when $\lambda \in [0, 1]$

| Metric | Value | Relative to mean(original) (%) |
|---|---|---|
| Mean absolute error (MAE) | 0.001 557 | 0.447 532 % |
| MAE (bootstrap mean) | 0.001 553 | 0.446 387 % |
| MAE 95% CI (bootstrap) | 0.000 934 − 0.002 268 | 0.268 478 − 0.651 936 % |
| Root mean squared error (RMSE) | 0.002 070 | 0.594 985 % |
| Median absolute error | 0.001 137 | 0.326 711 % |
| Bland–Altman mean difference (mean of $y - x$) | −0.000 399 | −0.114 728 % |
| Bland–Altman SD of differences | 0.002 103 | 0.604 310 % |
| Limits of agreement (mean $\pm$ 1.96 SD) | −0.004 520 − 0.003 722 | −1.299 176 − 1.069 720 % |

Notes: *Value* columns report absolute errors on the same scale as the original data. *Relative* column uses mean(original) = 0.347 938. Limits of agreement are computed as mean difference $\pm 1.96 \times$ SD.

**Table 9:** Summary error metrics for DeRaDiff approximations for CLIP scores when $\lambda > 1$

| Metric | Value | Relative to mean(original) (%) |
|---|---|---|
| Mean absolute error (MAE) | 0.007 215 | 2.073 730 % |
| MAE (bootstrap mean) | 0.007 302 | 2.098 770 % |
| MAE 95% CI (bootstrap) | 0.002 903 − 0.013 009 | 0.834 311 − 3.738 885 % |
| Root mean squared error (RMSE) | 0.012 594 | 3.619 573 % |
| Median absolute error | 0.002 391 | 0.687 314 % |
| Bland–Altman mean difference (mean of $y - x$) | −0.006 824 | −1.961 343 % |
| Bland–Altman SD of differences | 0.010 956 | 3.148 884 % |
| Limits of agreement (mean $\pm$ 1.96 SD) | −0.028 298 − 0.014 650 | −8.133 156 − 4.210 469 % |

Notes: *Value* columns report absolute errors on the same scale as the original data. *Relative* column uses mean(original) = 0.347 938. Limits of agreement are computed as mean difference $\pm 1.96 \times$ SD.

## A.10 STATISTICAL ANALYSIS OF DERADIFF'S PERFORMANCE ON HPS

### A.10.1 SDXL

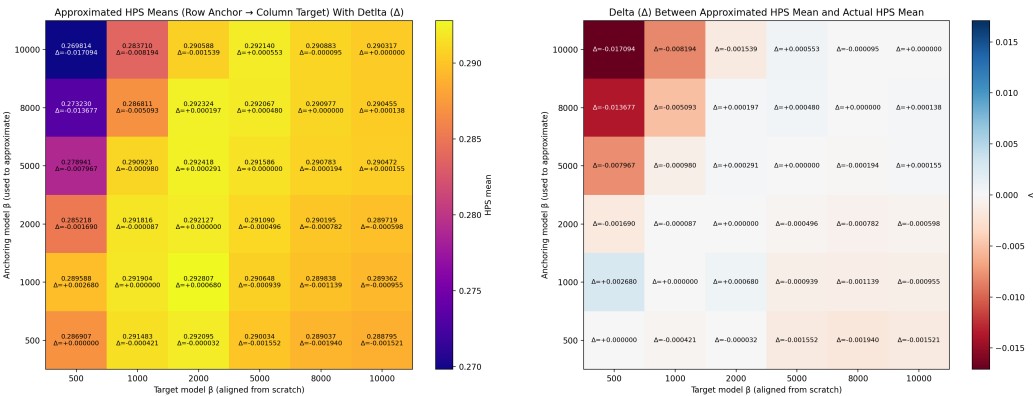

Figure 16: Approximated HPS Means with all (Row Anchor $\beta \rightarrow$ Column Target $\beta$) with Delta ($\Delta$)

Figure 17: Delta ($\Delta$) Between Approximated HPS Mean and Actual HPS Mean

**Table 10:** Summary error metrics for DeRaDiff approximations for HPS scores when $\lambda \in [0, 1]$

| Metric | Value | Relative to mean(original) (%) |
|---|---|---|
| Mean absolute error (MAE) | 0.000 770 | 0.264 808 % |
| MAE (bootstrap mean) | 0.000 770 | 0.265 098 % |
| MAE 95% CI (bootstrap) | 0.000 501 – 0.001 053 | 0.172 547 – 0.362 301 % |
| Root mean squared error (RMSE) | 0.000 949 | 0.326 627 % |
| Median absolute error | 0.000 680 | 0.233 938 % |
| Bland–Altman mean difference (mean of $y - x$) | −0.000 640 | −0.220 152 % |
| Bland–Altman SD of differences | 0.000 726 | 0.249 753 % |
| Limits of agreement (mean ± 1.96 SD) | −0.002 063 – 0.000 783 | −0.709 668 – 0.269 364 % |

Notes: *Value* columns report absolute errors on the same scale as the original data. *Relative* column uses mean(original) = 0.290 636. Limits of agreement are computed as mean difference ±1.96×SD.

**Table 11:** Summary error metrics for DeRaDiff approximations for HPS scores when $\lambda > 1$

| Metric | Value | Relative to mean(original) (%) |
|---|---|---|
| Mean absolute error (MAE) | 0.004 041 | 1.390 437 % |
| MAE (bootstrap mean) | 0.004 079 | 1.403 635 % |
| MAE 95% CI (bootstrap) | 0.001 746 – 0.006 886 | 0.600 884 – 2.369 320 % |
| Root mean squared error (RMSE) | 0.006 582 | 2.264 714 % |
| Median absolute error | 0.001 539 | 0.529 484 % |
| Bland–Altman mean difference (mean of $y - x$) | −0.003 481 | −1.197 693 % |
| Bland–Altman SD of differences | 0.005 782 | 1.989 560 % |
| Limits of agreement (mean ± 1.96 SD) | −0.014 814 – 0.007 853 | −5.097 232 – 2.701 845 % |

Notes: *Value* columns report absolute errors on the same scale as the original data. *Relative* column uses mean(original) = 0.290 636. Limits of agreement are computed as mean difference ±1.96×SD.

A.10.2  SD1.5

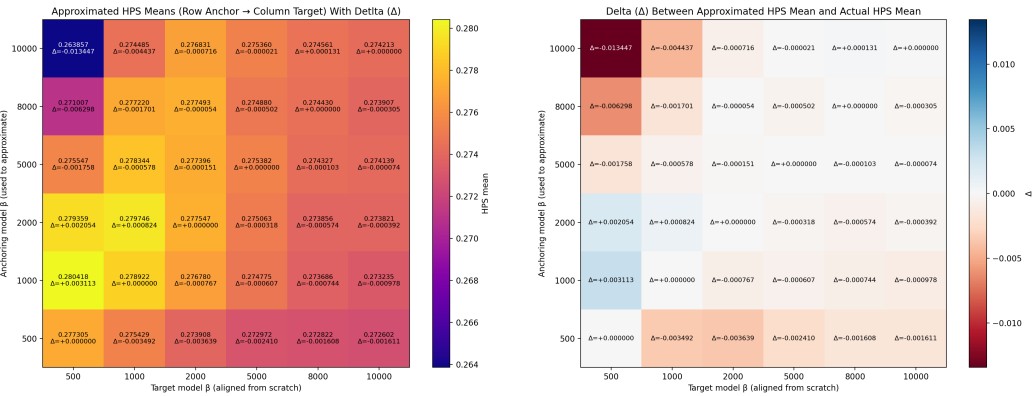

Figure 18: Approximated HPS Means with all (Row Anchor $\beta \rightarrow$ Column Target $\beta$) with Delta ($\Delta$)

Figure 19: Delta ($\Delta$) Between Approximated HPS Mean and Actual CLIP Mean

**Table 12:** Summary error metrics for DeRaDiff approximations for HPS scores when $\lambda \in [0, 1]$

| Metric | Value | Relative to mean(original) (%) |
|---|---|---|
| Mean absolute error (MAE) | 0.001 175 | 0.425 179 % |
| MAE (bootstrap mean) | 0.001 172 | 0.424 093 % |
| MAE 95% CI (bootstrap) | 0.000 654 − 0.001 781 | 0.236 744 − 0.644 486 % |
| Root mean squared error (RMSE) | 0.001 625 | 0.587 958 % |
| Median absolute error | 0.000 744 | 0.269 095 % |
| Bland–Altman mean difference (mean of $y - x$) | −0.001 175 | −0.425 179 % |
| Bland–Altman SD of differences | 0.001 161 | 0.420 353 % |
| Limits of agreement (mean $\pm$ 1.96 SD) | −0.003 451 − 0.001 102 | −1.249 071 − 0.398 713 % |

Notes: *Value* columns report absolute errors on the same scale as the original data. *Relative* column uses mean(original) = 0.276 300. Limits of agreement are computed as mean difference $\pm 1.96 \times$ SD.

**Table 13:** Summary error metrics for DeRaDiff approximations for HPS scores when $\lambda > 1$

| Metric | Value | Relative to mean(original) (%) |
|---|---|---|
| Mean absolute error (MAE) | 0.002 386 | 0.863 447 % |
| MAE (bootstrap mean) | 0.002 415 | 0.873 887 % |
| MAE 95% CI (bootstrap) | 0.000 986 − 0.004 310 | 0.356 733 − 1.560 056 % |
| Root mean squared error (RMSE) | 0.004 179 | 1.512 308 % |
| Median absolute error | 0.000 824 | 0.298 315 % |
| Bland–Altman mean difference (mean of $y - x$) | −0.001 569 | −0.567 982 % |
| Bland–Altman SD of differences | 0.004 009 | 1.450 790 % |
| Limits of agreement (mean $\pm$ 1.96 SD) | −0.009 426 − 0.006 287 | −3.411 531 − 2.275 566 % |

Notes: *Value* columns report absolute errors on the same scale as the original data. *Relative* column uses mean(original) = 0.276 300. Limits of agreement are computed as mean difference $\pm 1.96 \times$ SD.

## A.11 STATISTICAL ANALYSIS OF DERADIFF'S PERFORMANCE ON PICKSCORE

### A.11.1 SDXL

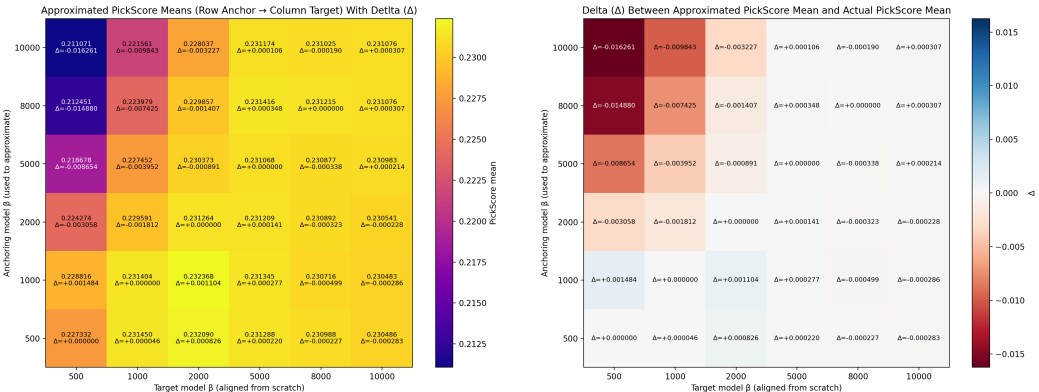

Figure 20: Approximated PickScore Means with all (Row Anchor $\beta \rightarrow$ Column Target $\beta$) with Delta ($\Delta$)

Figure 21: Delta ($\Delta$) Between Approximated PickScore Mean and Actual PickScore Mean

**Table 14:** Summary error metrics for DeRaDiff approximations for PickScore scores when $\lambda \in [0, 1]$

| Metric | Value | Relative to mean(original) (%) |
|---|---|---|
| Mean absolute error (MAE) | 0.000 355 | 0.153 864 % |
| MAE (bootstrap mean) | 0.000 353 | 0.153 119 % |
| MAE 95% CI (bootstrap) | 0.000 238 − 0.000 498 | 0.103 446 − 0.216 058 % |
| Root mean squared error (RMSE) | 0.000 441 | 0.191 478 % |
| Median absolute error | 0.000 283 | 0.122 888 % |
| Bland–Altman mean difference (mean of $y - x$) | 0.000 063 | 0.027 498 % |
| Bland–Altman SD of differences | 0.000 452 | 0.196 144 % |
| Limits of agreement (mean $\pm$ 1.96 SD) | −0.000 823 − 0.000 950 | −0.356 944 − 0.411 939 % |

Notes: *Value* columns report absolute errors on the same scale as the original data. *Relative* column uses mean(original) = 0.230 509. Limits of agreement are computed as mean difference $\pm 1.96 \times$ SD.

**Table 15:** Summary error metrics for DeRaDiff approximations for PickScore scores when $\lambda > 1$

| Metric | Value | Relative to mean(original) (%) |
|---|---|---|
| Mean absolute error (MAE) | 0.004 903 | 2.126 836 % |
| MAE (bootstrap mean) | 0.004 939 | 2.142 507 % |
| MAE 95% CI (bootstrap) | 0.002 544 − 0.007 666 | 1.103 610 − 3.325 670 % |
| Root mean squared error (RMSE) | 0.007 102 | 3.080 951 % |
| Median absolute error | 0.003 058 | 1.326 521 % |
| Bland–Altman mean difference (mean of $y - x$) | −0.004 644 | −2.014 716 % |
| Bland–Altman SD of differences | 0.005 562 | 2.412 729 % |
| Limits of agreement (mean $\pm$ 1.96 SD) | −0.015 545 − 0.006 257 | −6.743 664 − 2.714 232 % |

Notes: *Value* columns report absolute errors on the same scale as the original data. *Relative* column uses mean(original) = 0.373 395. Limits of agreement are computed as mean difference $\pm 1.96 \times$ SD.

### A.11.2 SD1.5

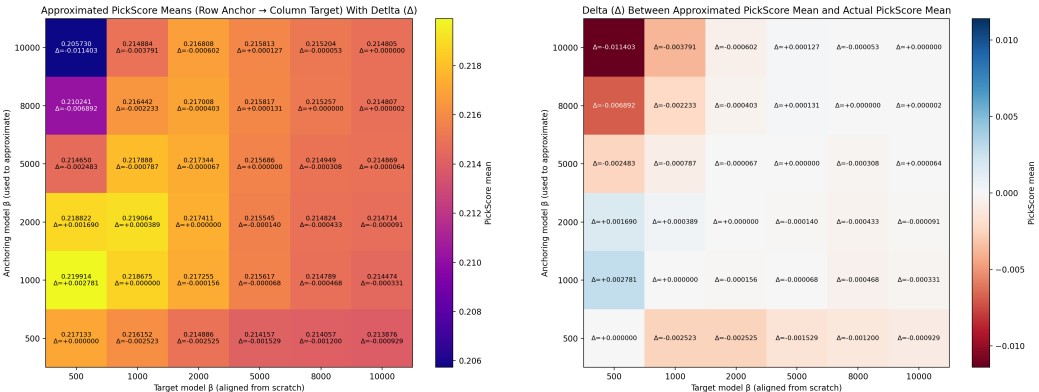

Figure 22: Approximated PickScore Means with all (Row Anchor $\beta \rightarrow$ Column Target $\beta$) with Delta ($\Delta$)

Figure 23: Delta ($\Delta$) Between Approximated PickScore Mean and Actual PickScore Mean

**Table 16:** Summary error metrics for DeRaDiff approximations for PickScore scores when $\lambda \in [0, 1]$

| Metric | Value | Relative to mean(original) (%) |
|---|---|---|
| Mean absolute error (MAE) | 0.000 718 | 0.331 616 % |
| MAE (bootstrap mean) | 0.000 717 | 0.331 069 % |
| MAE 95% CI (bootstrap) | 0.000 336 − 0.001 163 | 0.155 070 − 0.537 347 % |
| Root mean squared error (RMSE) | 0.001 097 | 0.506 768 % |
| Median absolute error | 0.000 331 | 0.153 080 % |
| Bland–Altman mean difference (mean of $y - x$) | −0.000 709 | −0.327 547 % |
| Bland–Altman SD of differences | 0.000 867 | 0.400 259 % |
| Limits of agreement (mean $\pm$ 1.96 SD) | −0.002 408 − 0.000 989 | −1.112 055 − 0.456 961 % |

Notes: *Value* columns report absolute errors on the same scale as the original data. *Relative* column uses mean(original) = 0.216 494. Limits of agreement are computed as mean difference $\pm 1.96 \times$ SD.

**Table 17:** Summary error metrics for DeRaDiff approximations for PickScore scores when $\lambda > 1$

| Metric | Value | Relative to mean(original) (%) |
|---|---|---|
| Mean absolute error (MAE) | 0.002 255 | 1.041 818 % |
| MAE (bootstrap mean) | 0.002 281 | 1.053 707 % |
| MAE 95% CI (bootstrap) | 0.000 973 − 0.003 959 | 0.449 230 − 1.828 517 % |
| Root mean squared error (RMSE) | 0.003 786 | 1.748 630 % |
| Median absolute error | 0.000 787 | 0.363 318 % |
| Bland–Altman mean difference (mean of $y - x$) | −0.001 573 | −0.726 613 % |
| Bland–Altman SD of differences | 0.003 564 | 1.646 340 % |
| Limits of agreement (mean $\pm$ 1.96 SD) | −0.008 559 − 0.005 413 | −3.953 440 − 2.500 213 % |

Notes: *Value* columns report absolute errors on the same scale as the original data. *Relative* column uses mean(original) = 0.216 494. Limits of agreement are computed as mean difference $\pm 1.96 \times$ SD.

### A.12 LLM USAGE

This research idea was conceived solely and only by the authors by identifying gaps in the existing research literature. LLMs were **NOT used** for any research ideation. LLMs were only used to polish writing, help in plotting graphs, retrieve known mathematical facts and fix any grammatical errors that the authors missed.

