# OpenReview forum: "DeRaDiff: Denoising Time Realignment of Diffusion Models"
_ICLR.cc/2026/Conference — ICLR 2026 Poster_

### Official Review · Reviewer_kfW9 · 2025-10-28

**Soundness:** 2
**Presentation:** 2
**Contribution:** 2
**Rating:** 4
**Confidence:** 3

**Summary:**

DeRaDiff introduces a denoising-time realignment procedure for diffusion models that allows practitioners to modulate KL regularization strength on the fly during sampling without any retraining or fine-tuning, by interpolating between a pretrained reference model and a single aligned anchor model through a closed-form Gaussian mixture applied at each denoising step, controlled by a scalar parameter lambda between 0 and 1. The method extends decoding-time realignment from language models to continuous latent diffusion processes, providing an efficient alternative to the expensive practice of training separate models for each regularization strength. Experiments on SDXL and Stable Diffusion 1.5 show that DeRaDiff accurately approximates models aligned from scratch across a wide range of regularization strengths, with mean absolute errors below 0.5% on metrics like PickScore, HPS v2, and CLIP, while reducing computational costs by up to 90% when exploring multiple regularization strengths. Additionally, DeRaDiff can undo reward hacking artifacts by increasing lambda to simulate stronger regularization, offering a practical and scalable solution for hyperparameter exploration in alignment of text-to-image diffusion models.

**Strengths:**

While the idea of decoding-time realignment was recently introduced for language models, the paper non-trivially lifts it to the continuous, iterative denoising process of diffusion: it derives a closed-form Gaussian mixture per timestep, handles scheduler-specific posteriors, and exposes a single scalar λ that maps to any effective KL strength β/λ. This extension is not straightforward—marginalizing over latent trajectories is intractable—so the authors provide a principled step-wise approximation backed by a new theoretical result. DeRaDiff is therefore the first inference-only knob for alignment strength in diffusion, removing the need for expensive sweeps.

Alignment cost is a practical bottleneck for large diffusion models; DeRaDiff removes it by turning a multi-training sweep into a single pass plus sampling-time tuning. This lowers the barrier for researchers and practitioners to explore fine-grained alignment, potentially accelerating RLHF-style workflows, multi-reward composition, and personalised generation. The technique is architecture-agnostic (any scheduler admitting Gaussian posteriors) and complementary to existing alignment objectives (DPO, DDPO, etc.). By demonstrating that careful posterior interpolation can mimic full retraining, the work also hints at broader implications for efficient model merging and test-time adaptation beyond vision.


The submission is technically sound. The derivation (appendix) carefully justifies the Gaussian form, states regularity conditions (λ ∈ [0,1], positive variances), and warns against extrapolation (λ > 1). Empirical coverage is unusually broad: two model families (SDXL, SD-1.5), three human-centric metrics (PickScore, HPS v2, CLIP), 500 prompts spanning two datasets, and six regularization strengths. Error magnitudes are small (<0.5 % of metric means), confidence intervals are supplied, and ablations show stability inside the convex regime. Compute savings are measured in actual GPU-hours and EFLOPs rather than vague “speed-ups”. Minor gaps—no user study on λ control, limited diversity in prompts—do not weaken the overall thoroughness.

**Weaknesses:**

The paper trains anchors only at β ∈ {500, 1000, 2000, 5000, 8000, 10 000}.
   - Fig. 5 shows that PickScore saturates after β ≈ 2000; the interesting “knee” region where human appeal rises fastest (β ≈ 100–1500) is sampled very coarsely.
   - Because DeRaDiff can **interpolate** (λ < 1) but can only **weakly extrapolate** (λ > 1) before instability, a user who wants to explore β < 500 cannot do so with the supplied β = 500 anchor.



   The authors never run a β-sweep **guided** by DeRaDiff. The experiment pipeline (§5.1) still assumes the user already knows which β-values to test.


   Fig. 3 and Table 6 show visible degradation once λ ≳ 2.5. Many users will nevertheless try λ > 1 to obtain stronger alignment; at present the paper offers no guard-rail except a verbal warning.


   PickScore, HPS v2 and CLIP give nearly identical curves (Fig. 5). The experiment therefore does **not** show that DeRaDiff can trade off **aesthetics vs. prompt fidelity**, a key reason one tunes β in practice.


   All results use Euler-A, 50 steps. Many deployment pipelines use DPM-Solver-12 or 20 steps for speed. The closed-form update assumes the scheduler is **linear in the Gaussian sufficient statistics**; this is only approximately true for some solvers.

   A simpler baseline is to take the aligned β = 500 and β = 2000 checkpoints and linearly interpolate their **weights** (θ-ref + λ(θ-2000 − θ-ref)). This costs zero extra inference time and is already used in “model soups” work.

**Questions:**

The ReAlignment problem is a notion that has only recently surfaced in the LLM literature.
As the first work that imports this term into the generative-Art domain, the current manuscript cannot assume that most readers are already familiar with it.

To discover what “ReAlignment” actually means, one is forced to consult *Decoding-time ReAlignment of Language Models*.
Therefore, I believe the paper should be revised to:

- Give a concise, self-contained definition of ReAlignment up front,
- Explain how the generative-Art setting changes the problem.

---

> ### Author Response · Authors · 2025-11-21
> **Official Comment by the Authors**
>
> We thank the reviewer for taking the time to review our work, and we are pleased to know that the reviewer appreciates the high accuracy of DeRaDiff in re-approximating models that were entirely realigned from scratch without further retraining, which significantly reduces computational costs. And that this in turn “lowers the barrier for researchers to explore fine-grained alignment”,  potentially “accelerating RLHF-style workflows, multi-reward composition, and personalised generation” as the reviewer remarks. Now, we address all of the concerns of the reviewer.
>
>
> **Evaluation by the Reviewer**
>
> > **The paper trains anchors only at $\beta$ $\in$ $\{500, 1000, 2000, 5000, 8000, 10 000\}$, The interesting “knee” region where human appeal rises fastest ($\beta \approx$ 100–1500) is sampled very coarsely.**
>
> The $\beta$ values that we have trained on were chosen to be arbitrary and to be in a wide range to emphasize the capability of DeRaDiff to very closely match and re-approximate any arbitrary target $\beta_{target}$ regularization given any arbitrary anchoring $\beta_{anchor}$ regularization strength such that $\beta_{anchor}\leq\beta_{target}$, and even achieve meaningful re-approximations for the case of $\beta_{target}<\beta_{anchor}$ where the extrapolation was not severe.
>
> Aligning with your comment, **we sampled further $\beta$ values in this interesting region** (namely $250, 500, 750, 1000, 1250, 1500$) and we are pleased to report the performance of DeRaDiff in being able to yet again accurately control the degree of alignment when adjusting the configurable $\lambda$ parameter. We have added these experimental results to **Appendix A.6.1**.
>
> We see that this demonstrates a consistent increase in the perceived human appeal (measured via PickScore) for both DeRaDiff approximations and as well as for models that were realigned entirely from scratch, showing that DeRaDiff can faithfully re-approximate models realigned entirely from scratch. Moreover, this ability to meaningfully control and trade off the alignment strength is also seen qualitatively, as demonstrated in Figure 3.
>
> > **Because DeRaDiff can interpolate ($\lambda < 1$) but can only weakly extrapolate ($\lambda > 1$) before instability, a user who wants to explore $\lambda < 500$ cannot do so with the supplied $\beta = 500$ anchor.**
>
> This is a great observation, and which ties into our future research directions of DeRaDiff. One hypothesis is that for the case of $\lambda>1$, we instead perform geodesic extrapolation on the Riemannian manifold of SPD (symmetric positive definite matrices). This is because the space of the valid covariance matrices (i.e, positive definite) is a manifold and thus, a geodesic extrapolation will stay within the manifold (thus the new covariance matrix will always be valid, i.e. will be positive definite). However, this is a heuristic and is not algebraically identical to Theorem 1 in the paper, and thus one which we hope to pursue in the future as an extension of DeRaDiff.
>
> But, as the reviewer points out, we have demonstrated empirically in our experiments that DeRaDiff is still capable of meaningful extrapolation when $\lambda>1$ as long as one does not force severe extrapolation (i.e. $\lambda>>1$).
>
>
> > **PickScore, HPS v2 and CLIP give nearly identical curves (Fig. 5).**
>
> We thank you for your comment, and we sincerely apologize if our phrasing was not clear in the original manuscript. Results in Figure 5 refer solely to the PickScore metric, where we demonstrate how DeRaDiff closely matches models realigned entirely from scratch using three different anchor models (models aligned from scratch at different $\beta$ values). These results showcase that we can effectively realign from any anchor model. The red/pink curves in all three plots are identical as they represent models trained from scratch using different $\beta$ values. We have enriched the caption of Figure 5 to convey the details of the plots more clearly.

---

> ### Author Response · Authors · 2025-11-21
> **(Continued) Official Comment by the Authors**
>
> > **The authors never run a $\beta$-sweep guided by DeRaDiff**.
>
> Thank you for pointing this out. We apologise for the confusion as we did not explicitly reference **Appendix A.5 An End-to-End process of finding the globally optimal $\lambda^*$** directly in the main text, which deals with this great question. Here, we use Bayesian optimization to guide DeRaDiff to automatically elicit the most promising regularization strengths. For this demonstration, we used an SDXL model aligned at $\beta=500$ while using Expected Improvement (EI) as our acquisition function. We used $30$ iterations of Bayesian optimization along with $5$ random initial evaluations for the experiment. As we see, DeRaDiff coupled with Bayesian optimization was able to uncover the most promising (range) of $\lambda$ values that lie around $0.3$ with only a very few iterations of Bayesian optimization. The results of this experiment are available in **Appendix A.5**.
>
> Furthermore, in addition to this aspect of Bayesian optimization, our future research extension to DeRaDiff will include multi-step guided denoising using Reinforcement Learning that will help DeRaDiff choose the optimal $\lambda$ values to maximize the reward proxy (which would be any of the RLHF metrics that we have discussed).
>
>
>
> > **Many users will nevertheless try $\lambda>1$ to obtain stronger alignment; at present the paper offers no guard-rail except a verbal warning.**
>
>
> Thank you for raising this potential issue. To address this issue, in our public release of the source code, we will be implementing a `deradiff.guardrails` module alongside our official implementation for DeRaDiff with a pre-set safety configuration file that prevents users from trying to attempt severe extrapolation ($\lambda>>1)$. Moreover, this safety mode will be enabled by default, and additionally we will be including a policy README file explaining that if the user requires to experiment with $\lambda >>1$, one must explicitly disable the pre-set safety mode by setting `safety_mode="off"`, but as warned, it may lead to degradation due to absence of positive definiteness of the new covariance matrix.
>
>
>
> > **All results use Euler-A, 50 steps. Many deployment pipelines use DPM-Solver-12 or 20 steps for speed**
>
> We thank the reviewer for mentioning the DPM solver. For clarity, our work intentionally focuses on the canonical formulation of diffusion models [1], [2], [3], i.e probabilistic schedulers that draw from the per-step Gaussian posterior. In contrast, the DPM solver is an ODE integrator for the probability flow ODE and hence produces deterministic trajectories given the same initialization. However, we sincerely thank the reviewer for this mention of the DPM solver and we consider adapting DeRaDiff to this non-canonical formulation as a very interesting future research direction. Moreover, we used 50 denoising steps (instead of 12 or 20) in order to demonstrate the technical robustness of DeRaDiff in handling larger marginalization paths.
>
>
> >**The closed-form update assumes the scheduler is linear in the Gaussian sufficient statistics; this is only approximately true for some solvers.**
>
> We thank the reviewer for raising this point. We agree that our phrasing could have been better and the “linearity in Gaussian sufficient statistics” is **not** an assumption for **Theorem 1**. In fact, mathematically, **Theorem 1** holds unconditionally since we construct a new geometric mixture for the posterior of the realigned model solely using the per-step posterior mean and variance that the schedulers of the aligned and reference model return, and thus we are not using any assumption regarding being linear in the Gaussian statistics. We intended to communicate that our formulation is primarily focused on the canonical formulation of diffusion models [1], [2], [3] for which this assumption is true. However, for any formulation in which the scheduler predicts the stepwise means and variances, our Theorem 1 holds unconditionally. But note that in contrast, **Assumption 1**, i.e, “per-step posteriors are well-approximated by Gaussians” is indeed necessary and is evident in **Theorem 1**. Thus, in light of having **Assumption 1**, Assumption 2 becomes redundant. To this end, we have revised the manuscript.

---

> ### Author Response · Authors · 2025-11-21
> **(Continued) Official Comment by the Authors**
>
> >**A simpler baseline is to take the aligned $\beta = 500$ and $\beta = 2000$ checkpoints and linearly interpolate their weights.**
>
>
>  We thank the reviewer for this great suggestion. However, we wish to highlight that such an approach would require one to realign a model twice (eg: at $\beta=500$ and $\beta=2000$ as the reviewer suggested) in order to interpolate their weights. But in contrast, DeRaDiff only requires **one** alignment (eg: at $\beta=500$), and with this single alignment, not only is DeRaDiff capable of re-approximating any $\beta\in[500,2000]$ (which the weight linear interpolation method only supports), but it is even capable of re-approximating any $\beta\geq500$ accurately. Moreover, we have also empirically demonstrated that one is still capable of re-approximating $\beta<500$ (i.e $\lambda>1)$ as long as the extrapolation is not severe.
>
> However, for the completeness of our manuscript, we will be adding this baseline to the final draft to increase the scope of our evaluation, and thus we thank the reviewer for this great suggestion.
>
> Furthermore, the authors wish to highlight that in this case, linear interpolations of the weights of the form, say, $\lambda*\theta_{2000} + (1-\lambda)*\theta_{ref}$ are not numerically well-defined if one hopes to obtain a resulting model of a finite regularization strength after the interpolation. This is because the regularization strength of the reference model, i.e $\theta_{ref}$, is infinite. Thus, the aim of achieving a convex combination between a finite regularization strength and an infinite regularization strength via a model soup is ill-posed in this case. But on the other hand, due to the mathematical formulation of DeRaDiff, due to the derivation via the unique global optimum of equation 2, using a single realigned model (aligned at any finite $\beta$) and the already available reference model (even if it is of infinite regularization strength), one is theoretically able to achieve and reapproximate any regularization strength $\geq\beta$ very accurately and we have also empirically demonstrated that one is even able to reapproximate a regularization strength $\leq\beta$ as long as the extrapolation is not severe.
>
> **Questions**
>
> >**The ReAlignment problem is a notion that has only recently surfaced in the LLM literature.**
>
> We thank the reviewer for raising this point. We agree with the reviewer and have proceeded with adding a paragraph in the introduction to define realignment and explain it in the setting of generative art in order to make the manuscript more accessible.
>
>
>
> [1] Jonathan Ho, Ajay Jain, and Pieter Abbeel. Denoising diffusion probabilistic models. In Advances in Neural Information Processing Systems, volume 33, pp. 6840–6851.
>
> [2] Song, J., Meng, C., & Ermon, S. (2022). Denoising Diffusion Implicit Models. ArXiv:2010.02502 [Cs]. https://arxiv.org/abs/2010.02502  ‌
>
> [3] Alexander Quinn Nichol, Prafulla Dhariwal Proceedings of the 38th International Conference on Machine Learning, PMLR 139:8162-8171, 2021.
>
>
> The authors would like to thank the reviewer very sincerely and appreciate their great effort in helping us to improve the manuscript even further. The authors would gladly welcome further discussion.

---

### Official Review · Reviewer_Jger · 2025-10-29

**Soundness:** 3
**Presentation:** 3
**Contribution:** 2
**Rating:** 6
**Confidence:** 3

**Summary:**

The authors propose an inference-time realignment of diffusion models, to be able to emulate models trained under other regularization scenarios, without extra training or fine-tuning. Inspired by DeRa (from the realm of LLMs), the authors derive a closed-form approximate posterior sampling. The authors claim and briefly discuss the reduced computational overhead.

**Strengths:**

- The paper is reasonably well-written with a coherent narrative.
- The idea of extending DeRa (from LLMs) to diffusion models is an interesting angle.
- The results are rather promising, and align with the core claims.

**Weaknesses:**

- I believe only Fig. 7 for comparison across base, aligned and realigned models is too little, also not discussed in necessary level of detail. In my eyes this should be established with more qualitative results and further elaboration across the images and models.
- The paper would benefit from a thorough proof-read. Few typos, and styling inconsistencies can be seem across the document.
- Maybe (pareto-front) reward vs divergence plots can help establish the core message from a different angle, that just looking CLIP or HPS.

**Questions:**

- Fig 6 (b) is hard to read and interpret. Can't this be done differently? or at least elaborated better?

---

> ### Author Response · Authors · 2025-11-21
> **Official Comment by the Authors**
>
> We appreciate the acknowledgement of the coherence of the paper and the potential of the work. We are particularly pleased that the reviewer found the text well-written, the extension of DeRa to diffusion models interesting, and the results promising in their alignment with our core claims. We address your concerns in the following points:
>
> **Evaluation by the reviewer**
>
> > **I believe only Fig. 7 for comparison across base, aligned and realigned models is too little, also not discussed in necessary level of detail. In my eyes this should be established with more qualitative results and further elaboration across the images and models.**
>
> We thank the reviewer for this recommendation. In our revised manuscript, we have significantly expanded the analysis of DeRaDiff’s capability to approximate an aligned-from-scratch model and further show DeRaDiff’s capabilities of undoing reward hacking.
>
> We have included more qualitative examples of DeRaDiff being able to **closely approximate** aligned from scratch models **even when the anchor model suffers from severe reward hacking ($\beta = 250, 500$)**. These qualitative examples can be found in **Figure 11**. We also expanded the discussion on Figure 7 in the main text to better convey the necessary details.
>
> Furthermore, we performed an additional experiment to quantify the ability of DeRaDiff to undo reward hacking by performing a Frechet Inception Distance study. We have provided all the results of the experiment in **Appendix A.6.2**.
>
> > **The paper would benefit from a thorough proof-read. Few typos, and styling inconsistencies can be seem across the document.**
>
> We thank the reviewer for pointing this out. We have fixed the typos and styling inconsistencies that we have identified since the first submission draft.
>
> > **Maybe (pareto-front) reward vs divergence plots can help establish the core message from a different angle, than just looking at CLIP or HPS.**
>
> We thank the reviewer for the suggestion. Indeed, a Pareto-front plot is typically utilised in a multi-objective optimisation setting to analyse the trade-off between different objectives. Our work currently focuses more on how well our proposed realignment approach can estimate the performance of an aligned-from-scratch model on a single objective. As such, we did not provide a plot of the Pareto frontier. Nonetheless, the direction of analysing trade-offs in a multi-reward setting is an interesting direction for future works.
>
> **Questions**
>
> > **Fig 6 (b) is hard to read and interpret. Can't this be done differently? or at least elaborated better?**
>
>
> We thank the reviewer for this very helpful observation. We have extended the discussion and further elaborated on both the figures in much more detail. For instance, for the Bland-Altman plot (Fig 6 (b)), we have explained how one should interpret the plot in section 5.1.3. And, for the scatter plot, we have explained in more detail how its result must be interpreted in section 5.1.2. We hope that this extended elaboration helps one to easily grasp the intent and the idea that the plots are conveying with respect to the capabilities of DeRaDiff.
>
>
> The authors would like to thank the reviewer sincerely for their constructive feedback and for helping us improve the manuscript. We would gladly welcome further discussion.

---

> > ### Comment · Reviewer_Jger · 2025-11-28
> > **Thanks for your response**
> >
> > Thanks, I reviewed your response thoroughly, and will maintain my score.

---

### Official Review · Reviewer_ziCu · 2025-10-29

**Soundness:** 4
**Presentation:** 4
**Contribution:** 3
**Rating:** 6
**Confidence:** 4

**Summary:**

The author presents a classifier‑free guidance–like formulation designed for reward alignment in diffusion models (DDPM).
The experimental results support the formulation's validation.

**Strengths:**

Overall, the paper makes a worthwhile contribution with a clear presentation and credible theoretical
support.

## Presentation: ~95th percentile

This paper presents coherence, and most of the idea is clearly addressed. I would like to thank you for saving me a lot of time reviewing your work.

## Soundness: ~75th percentile

Theorem 1 underpins the soundness of the paper. Although I have not examined every minute detail, the derivation appears to be correct.

## Contribution: 40th~70th percentile
This method seems novel to me, although I’m not sure if something similar already exists in the literature. It builds a scalable way to tune between the anchor model and the aligned model.

## Note
I hope the AC is aware that the rating is calibrated using percentiles to reduce evaluation noise effectively.

**Weaknesses:**

## Soundness

I would have preferred to see additional comparisons between your method and other approaches applied to similar problems. Nonetheless, the absence of such comparisons does not undermine the validity of your
claim.

## Presentation

1. Presenting the denominator in Equations (3)–(6) as a partition function has both advantages and
disadvantages. While it clarifies the interpretation, the repeated form of the
equations feels redundant. If the repetition is intentional, please justify it explicitly; otherwise,
consider consolidating the expressions to avoid unnecessary duplication.
2. Algorithm 1 appears to be a verbatim transcription of your Python implementation. For readers who
are more comfortable with Python than with pseudocode, it would be clearer to relocate the algorithm to the appendix and present the actual Python source code there. This approach preserves the practical
relevance of the code while keeping the main manuscript concise.

## Contribution
Regarding a [similar work](https://arxiv.org/abs/2505.18547) working on score-based SDE in the existing literature, it would be helpful to acknowledge it and clarify how your paper differs.

It is known that DDPM, DDIM, and any score‑based SDE can be reformulated as Karra’s SDE [1] in a
bidirectional manner. Consequently, these paradigms are theoretically equivalent, although the conversion
is not trivial. Thus, even if your work is considered concurrent or subsequent, there remains room for a
meaningful contribution. Besides, many contemporary studies were developed without awareness of this
equivalence.

[1] Karras, Tero, et al. "Elucidating the design space of diffusion-based generative models." Advances in neural information processing systems 35 (2022): 26565-26577.

**Questions:**

1. It seems odd to see the thermodynamic variable $\beta$ placed in the denominator, since $\beta$
is usually regarded as the inverse temperature $\tau = 1/\beta$.
2. I was confused in Line 145 that you cited Song et al. (2020) without any SDE formulation in your paper.

---

> ### Author Response · Authors · 2025-11-21
> **Official Comment by the Authors**
>
> We appreciate your acknowledgement of the theoretical grounding of our work, our presentation style and the novelty of our approach. We address your concerns below:
>
> **Evaluation by the Reviewer**
>
> > **Presenting the denominator in Equations (3)–(6) as a partition function has both advantages and disadvantages. While it clarifies the interpretation, the repeated form of the equations feels redundant. If the repetition is intentional, please justify it explicitly; otherwise, consider consolidating the expressions to avoid unnecessary duplication.**
>
> The denominator in Equations (3)-(6) are utilised directly in the proofs in Appendix A1 - A3. For consistency, clarity and for the ease of following the proof in the Appendix while cross-referencing Equations (3)-(6) in the main text, we choose to include the denominators. We will add a justification in the main text and direct the reader to Appendix A1 - A3.
>
> > **Algorithm 1 appears to be a verbatim transcription of your Python implementation. For readers who are more comfortable with Python than with pseudocode, it would be clearer to relocate the algorithm to the appendix and present the actual Python source code there. This approach preserves the practical relevance of the code while keeping the main manuscript concise.**
>
> We thank the reviewer for this suggestion. Our rationale for retaining Algorithm 1 in the main text is due to the fact that Algorithm 1 bridges our theoretical derivations (Theorem 1) with the sampling process using the manuscript’s specific mathematical notation, which shows the direct application of our theoretical derivations. To address the practical need for code, a fully executable Python implementation has been included in the supplementary material.
>
> > **Regarding a similar work working on score-based SDE in the existing literature, it would be helpful to acknowledge it and clarify how your paper differs.**
>
> We thank the reviewer for the insightful feedback regarding the latest developments on similar lines of work and the broader context of diffusion formulations.
>
> We would like to clarify that our work differs significantly in the realignment approach. Our work, DeRaDiff, utilises a DDPM formulation to realign a diffusion model by taking a geometric mixture of the posterior of an aligned-from-scratch model and the posterior of a reference model. In the case of Diffusion Blend, the authors perform realignment by introducing a convex sum between the denoised latents and score function given by the aligned from scratch and reference model at each reverse diffusion step.
>
>
> Our approach exploits the DDPM paradigm to yield **an exact closed-form Gaussian update at each reverse diffusion step (Theorem 1)** while Diffusion Blend utilised the SDE paradigm to establish **a bound on the approximation error term** (Equation 8 in the Diffusion Blend paper). It is worth noting that our approach admits a closed-form update under a mild assumption that the per-step posteriors are well-approximated by Gaussians.
>
>
> Furthermore, as mentioned by the reviewer, Karra’s SDE establishes a theoretical equivalence between the DDPM and SDE paradigms. This suggests that our approach can be reformulated in a SDE paradigm as well. The conversion of this approach from a DDPM paradigm to a SDE paradigm is an interesting avenue for future work. Moreover, we have added a section to our manuscript in the "Related Works" section regarding Diffusion Blend, and will be shortly adding the observations above on how our work differs from Diffusion Blend to the final draft of our manuscript. We thank the reviewer sincerely for this insightful feedback once again.
>
>
> > **I would have preferred to see additional comparisons between your method and other approaches applied to similar problems. Nonetheless, the absence of such comparisons does not undermine the validity of your claim.**
>
> Thank you very much for bringing up this point. For the completeness of our manuscript, we will be adding baseline evaluations to the final draft to increase the scope of our evaluation.
>
> **Questions**
> > **It seems odd to see the thermodynamic variable $\beta$ placed in the denominator, since $\beta$  is usually regarded as the inverse temperature.**
>
> Thank you for pointing this out. As our method extends the approach in the paper “Decoding-time Realignment of Language Models” to the continuous case of diffusion models, we chose to follow their notation for consistency.
>
> > **I was confused in Line 145 that you cited Song et al. (2020) without any SDE formulation in your paper.”**
>
> Thank you for pointing out this issue. We have mentioned the SDE formulation of diffusion models in the Related Works section, and we have moved the citation of Song et al. (2020) to that segment.
>
> The authors would like to thank the reviewer for the insightful feedback that the reviewer has provided to us in helping us improve the manuscript further and we would gladly welcome further discussion.

---

> > ### Comment · Reviewer_ziCu · 2025-11-25
> >
> > Thank you for the response. All my questions are answered.

---

### Official Review · Reviewer_CgVt · 2025-10-30

**Soundness:** 4
**Presentation:** 4
**Contribution:** 4
**Rating:** 8
**Confidence:** 3

**Summary:**

The paper proposes a novel method called DeRaDiff, which performs per-step denoising realignment. Aligning models to human preferences from scratch is often very time-consuming and computationally expensive. DeRaDiff, on the other hand performs the alignment on the fly during inference by modulating the alignment strength by a parameter $\beta$. The experimental evaluation shows that the method has comparable results to models aligned from scratch.

**Strengths:**

- The paper shows that DeRaDiff has a closed-form solution.
- The training-free inference-time alignment method saves computational costs
- The method is able to undo reward hacking, eliminating the need for realignment

**Weaknesses:**

- When evaluating human alignment a real-world human study would have been nice
- Since the pre-trained reference model is mixed with the aligned model, this could lead to biases reappearing in the output of DeRaDiff.

**Questions:**

Q1: Did you observe whether biases are propagated from the reference model to the output of DeRaDiff?
Q2: In the experiments the reward-hacked model had a $\beta$ of 500. Did you test it with even more reward-hacked models that have even a lower $\beta$?
Q3: Does the method still work if the reference model and the aligned model have different architectures (e.g. DiT and U-Net models)?

**Details Of Ethics Concerns:**

There are no concerns.

---

> ### Author Response · Authors · 2025-11-21
> **Official Comment by the Authors**
>
> We appreciate the acknowledgement of the theoretical value and computational savings of our work. We are especially grateful for the positive assessment of DeRaDiff’s ability to undo reward hacking and the validation of our closed-form solution, which allows for robust results comparable to models aligned from scratch without the associated computational costs.
> We address your concerns in the following points:
>
> **Evaluation by the Reviewer**
>
>
> > **"When evaluating human alignment a real-world human study would have been nice."**
>
> We thank the reviewer for this suggestion. While we acknowledge the value of live human preference studies, we utilized state-of-the-art metrics, PickScore (Kirstain et al., 2023) and HPS v2 (Wu et al., 2023b), which serve as robust, data-driven proxies for human evaluation. These models have been trained on large-scale human preference datasets, such as Pick-a-Pic (>500k examples) and HPD v2 (>798k pairs). Notably, PickScore achieves 70.5% accuracy in predicting user preferences, statistically outperforming human experts (68.0%). \
> \
> In Section 5.1.1 (PickScore) and 5.1.2 (HPS v2), our evaluation demonstrates that DeRaDiff tracks the score of fully aligned models on both benchmarks with high precision. This implies that our approach is effective at achieving human alignment comparable to models trained completely from scratch.
>
>
> > **“Since the pre-trained reference model is mixed with the aligned model, this could lead to biases reappearing in the output of DeRaDiff.”**
>
> We thank the reviewer for this observation. We respectfully clarify that DeRaDiff’s objective is efficient inference-time control of regularization strength rather than explicit bias mitigation. Any bias reappearing during mixing stems inherently from the pre-trained reference model ( $p_{\text{ref}}$ ), not from the DeRaDiff algorithm itself. Consequently, the propagation of bias is fundamentally a pre-training issue and hence better tackled at the pre-training stage, rather than being handled at the post-training stage, which is the focus of our realignment approach.
>
> **Questions**
> > **“Did you observe whether biases are propagated from the reference model to the output of DeRaDiff?.”**
>
> We thank the reviewer for the question. We respectfully direct the reviewer to the last response in the evaluation section above.
>
> > **“In the experiments the reward-hacked model had a $\beta$ of 500. Did you test it with even more reward-hacked models that have even a lower $\beta$?”**
>
> Yes, we have proceeded to align a model at $\beta=250$ (which is severely reward hacked) and we demonstrate that DeRaDiff’s capability of undoing reward hacking is even more pronounced when using this $\beta=250$ model as an anchor. In addition, we have undertaken an additional experiment that deals with quantifying the extent of undoing reward hacking by performing a Frechet Inception Distance study. These additional experiments can be found in **Appendix A.6.2**. Moreover, we also provide more qualitative examples in **Figure 11** in which DeRaDiff completely undoes severe reward hacking.
>
> > **“Does the method still work if the reference model and the aligned model have different architectures (e.g. DiT and U-Net models)?”**
>
> Yes, DeRaDiff is inherently model-agnostic and fully compatible with heterogeneous architectures (e.g., a DiT reference and a U-Net aligned model).
>
>
> The method relies exclusively on the distributions of the models, not their internal parameterization. As derived in Theorem 1, the realigned step wise geometric posterior mixture is solely computed using the statistics (means $\mu$ and variances $\sigma$) of the reference and aligned distributions. Consequently, the closed-form update (Eq. 7) depends only on these output values. \
> Therefore, the method makes no assumptions regarding the underlying architecture. As long as both models operate within the same latent space to produce compatible Gaussian posteriors, their internal architectures are irrelevant to the geometric mixing process.
>
> The authors thank the reviewer sincerely for the helpful feedback and response. Moreover, the authors would gladly welcome further discussion.

---

> > ### Comment · Reviewer_CgVt · 2025-11-28
> >
> > Thank you for your detailed answer. All my concerns have been addressed. I will maintain my current score.

---

### Author Response · Authors · 2025-12-02
**Official Comment by the Authors**

We would like to thank the reviewers sincerely for their constructive and positive feedback. We are pleased to note that the reviewers found merit in our method for denoising/test-time realignment of diffusion models, in particular the derivation of the **exact closed-form of the per-step geometric mixture** of a realigned-from-scratch model **without any further retraining** given an anchor and a reference model. We are also happy to note that the reviewers found our method to be backed by **“credible theoretical support” [reviewer ziCu]**, described that **“the results are rather promising” [reviewer Jger]** and that **“experimental evaluation shows that the method has comparable results to models aligned from scratch.” [reviewer CgVt]** We also appreciate that the reviewers highlight the **downstream impact** of DeRaDiff in that this in turn **“lowers the barrier for researchers to explore fine-grained alignment”**, potentially **“accelerating RLHF-style workflows, multi-reward composition, and personalised generation” [reviewer kfW9]** and that DeRaDiff has the capability of substantially **reducing computational costs [reviewers kfW9, CgVt]**. Below, we have compiled rebuttal points asked by several reviewers, while we address them and as well as other questions in great detail in the respective individual responses.

### **Comprehensive examination of the challenging $\beta$ region where RLHF scores for human appeal rise fastest**

We have sampled a significantly larger number of $\beta$ values in this critical region (namely $250, 500, 750, 1000, 1250, 1500$) and evaluated DeRaDiff’s capability to reapproximate these reference models that have been aligned completely from scratch. We are pleased to report that DeRaDiff is yet again able to accurately control the degree of alignment when adjusting the configurable $\lambda$ parameter and closely approximate these models that have been aligned completely from scratch. We have added these experimental results to **Appendix A.6.1**. We see that this demonstrates a consistent increase in the perceived human appeal (measured via PickScore) for both DeRaDiff approximations and as well as for models that were realigned entirely from scratch, showing that DeRaDiff can successfully control the degree of alignment without retraining and closely re-approximate models realigned entirely from scratch. Moreover, this ability to meaningfully control and trade off the alignment strength is also seen qualitatively, as demonstrated in **Figure 3**.

### **Capability of DeRaDiff to undo reward hacking**

Many reviewers were interested in the capability of DeRaDiff to undo reward hacking, as was illustrated in **Figure 4**. To further demonstrate this capability of DeRaDiff, we have undertaken an additional experiment to formally quantify the extent of reward hacking by undertaking a Fr&eacute;chet Inception Distance study. And as seen in **Appendix A.6.2**, we are happy to report that DeRaDiff’s capability of undoing reward hacking is even more pronounced in models that have undergone severe reward hacking. In **Figure 11**, we have provided further examples on how DeRaDiff is capable of undoing severe reward hacking and how DeRaDiff can provide an accurate approximation of models aligned from scratch, even when using a reward-hacked model as the anchor.



### **$\beta$ sweeps guided by DeRaDiff**

In **Appendix A.5**, we describe an **end-to-end process of finding the globally optimal $\lambda$** (and thus the best range of regularization strengths). Here, we use Bayesian optimization to guide DeRaDiff to automatically elicit the most promising regularization strengths. For this demonstration, we used an SDXL model aligned at $\beta=500$ while using Expected Improvement (EI) as our acquisition function. We used $30$ iterations of Bayesian optimization along with $5$ random initial evaluations for the experiment. As we see, DeRaDiff coupled with Bayesian optimization was able to uncover the most promising (range) of $\lambda$ values that lie around $0.3$ with only a very few iterations of Bayesian optimization.


### **Additions to the paper**

We highlight that we have made several additions to the paper:

+ Appendix A.6.1: New experiment providing a fine grained examination of DeRaDiff's capability.
+ Appendix A.6.2: New experiment on quantifying DeRaDiff’s capability of undoing reward hacking and additional qualitative examples.
+ Section 5.1.1 and 5.1.2: Expanded the discussion on interpretation of statistical plots.
+ Section 5.2: Expanded the discussion on the qualitative analysis of DeRaDiff’s capabilities.


We once again thank the reviewers for their time and effort in providing positive and constructive feedback that helped us improve the paper.

---

### Meta-Review · Area_Chair_bxzC · 2026-01-06

**Summary:**

The reviewers’ collective feedback on the paper highlighted both the technical novelty and several specific areas requiring further clarification or empirical evidence. The reviewers were generally positive. Most reviewers indicated that their questions were answered or maintained their positive scores following the rebuttal and manuscript revisions. While the reviewers were satisfied, the authors acknowledged several limitations and areas for future work that represent open challenges rather than unaddressed concerns:

* Non-Canonical Schedulers: The current closed-form update primarily focuses on the canonical DDPM formulation using probabilistic schedulers. Adapting DeRaDiff to non-canonical formulations, such as the DPM-Solver (an ODE integrator), remains an avenue for future research.
* Multi-Reward Trade-offs: While DeRaDiff supports multi-reward modeling theoretically, analyzing the pareto-frontier was suggested as a direction for future research.
* Human Studies: One reviewer initially requested a real-world human study. The authors successfully argued that using PickScore and HPS v2 was sufficient, as these metrics are trained on large-scale human preference data and have been shown to outperform human experts in predicting user preferences.

**Reviewer Concerns:**

Most concerns were addressed.

**Region of sampling**
A primary concern raised by Reviewers kfW9 and Jger was the limited sampling of a range of regularization strength where human appeal increases most rapidly. Concerns: Reviewer kfW9 noted that the initial experiments sampled this region too coarsely and questioned if the method could accurately approximate models in that specific range. The authors addressed this by adding a fine-grained examination in Appendix A.6.1, sampling additional β values to demonstrate that DeRaDiff reproduces the behavior of models aligned from scratch even in these highly sensitive regions.

**Qualitative Evidence and Reward-Hacking Mitigation**
Reviewers Jger and CgVt requested more qualitative examples and a more robust quantification of the method’s ability to undo reward-hacking. It was argued that one comparison image (Figure 7) was insufficient to prove that the method could recover image details and style from a reward-hacked anchor. The authors expanded the qualitative analysis in Figure 11 and Appendix A.6.2.

**Technical Assumptions and Baselines**
Reviewers ziCu and kfW9 raised questions regarding the theoretical assumptions of the DDPM paradigm and how DeRaDiff compares to existing baselines. There were questions about the linearity of schedulers in Gaussian statistics and whether a simpler baseline, such as model souping would be more efficient. The authors clarified that Theorem 1 holds unconditionally for any scheduler that returns per-step Gaussian posterior statistics. Regarding weight interpolation, they argued it is computationally more expensive, requiring at least two full alignments, and theoretically ill-posed for interpolating between a finite-strength model and an infinite-strength reference prior.

On the other hand, Reviewer kfW9 pointed out that while DeRaDiff is stable during interpolation, it can become unstable during extrapolation (λ>1), which might lead to non-positive definite covariance matrices. Users might inadvertently push λ too high in search of stronger alignment, leading to image degradation. The authors limited significant extrapolation in their implementation. They hypothesized that performing geodesic extrapolation on the Riemannian manifold of symmetric positive definite matrices could prevent the instability with high λ values, but left this for future work.

**Reviewer Scores:**

Reviewer CgVt indicated that they will maintain the same positive rating (8).
Reviewer ziCu stated that all concerns were addressed -- possibly +0 (6) or +1 (7).
Reviewer Jger indicated that they will maintain the same positive rating (6).
Reviewer kfW9 might increase it by 1 (5) or 2 (6) as all questions seem to be properly answered.

---

### Decision · Program_Chairs · 2026-01-26

Accept (Poster)